# Gaussian Process Upper Confidence Bound Achieves Nearly-Optimal Regret in Noise-Free Gaussian Process Bandits

**Shogo Iwazaki**
LY Corporation
Tokyo, Japan
siwazaki@lycorp.co.jp

## Abstract

We study the noise-free Gaussian Process (GP) bandit problem, in which a learner seeks to minimize regret through noise-free observations of a black-box objective function that lies in a known reproducing kernel Hilbert space (RKHS). The Gaussian Process Upper Confidence Bound (GP-UCB) algorithm is a well-known approach for GP bandits, where query points are adaptively selected based on the GP-based upper confidence bound score. While several existing works have reported the practical success of GP-UCB, its theoretical performance remains sub-optimal. However, GP-UCB often empirically outperforms other nearly-optimal noise-free algorithms that use non-adaptive sampling schemes. This paper resolves the gap between theoretical and empirical performance by establishing a nearly-optimal regret upper bound for noise-free GP-UCB. Specifically, our analysis provides the first constant cumulative regret bounds in the noise-free setting for both the squared exponential kernel and the Matérn kernel with some degree of smoothness.

## 1 Introduction

This paper studies the noise-free Gaussian Process (GP) bandit problem, where the learner seeks to minimize regret through noise-free observations of the black-box objective function. Several existing works tackle this problem, and some of them [Iwazaki and Takeno, 2025a, Salgia et al., 2024] propose algorithms whose regret nearly matches the lower bound of [Li and Scarlett, 2024]. For ease of theoretical analysis, these algorithms rely on the non-adaptive sampling scheme, whose query points are chosen independently of the observed function values, such as the uniform sampling [Salgia et al., 2024] or maximum variance reduction [Iwazaki and Takeno, 2025a]. Although the theoretical superiority of such non-adaptive algorithms is shown, in existing noisy setting literature [Bogunovic et al., 2022, Iwazaki and Takeno, 2025b, Li and Scarlett, 2022], their empirical performance has been reported to be worse than that of fully adaptive strategies such as GP upper confidence bound (GP-UCB) [Srinivas et al., 2010]. Unsurprisingly, we also observe such empirical and practical gaps in a noise-free setting, as shown in Figure 1. These observations suggest the possibility of further theoretical improvement in the practical fully adaptive algorithm. From this motivation, our work aims to establish the nearly-optimal regret for GP-UCB, which is one of the well-known adaptive GP bandit algorithms, and its existing guarantees only show strictly sub-optimal regret in a noise-free setting [Kim and Sanz-Alonso, 2024, Lyu et al., 2019].

**Contributions.** Our contributions are summarized below:

- We give a refined regret analysis of GP-UCB (Theorems 1 and 2), which matches both the cumulative regret lower bounds of [Li and Scarlett, 2024] and the simple regret lower bound

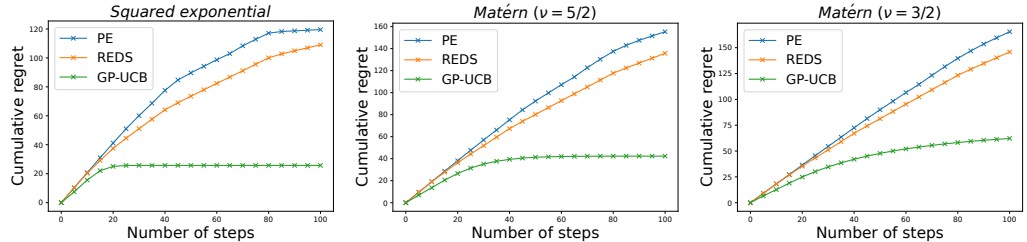

Figure 1: Empirical performance comparison of GP-UCB and two existing nearly-optimal algorithms: random exploration with domain shrinking (REDS) [Salgia et al., 2024] and phased elimination (PE) [Iwazaki and Takeno, 2025a]. From left to right, the plots show the average cumulative regret over 3000 independent runs under the squared exponential kernel, Matérn kernel ($\nu = 5/2$), and Matérn kernel ($\nu = 3/2$) with $d = 2$, respectively. Here, $d$ and $\nu$ represent the dimension of the input and the smoothness parameter, respectively. Detailed experimental settings are provided in Appendix B.

    of [Bull, 2011] up to polylogarithmic factors in the Matérn kernel. Regarding cumulative regret, our analysis shows that GP-UCB achieves the constant $O(1)$ regret under the squared exponential and Matérn kernel with $d > \nu$. The results are summarized in Tables 1 and 2.

- Our key theoretical contribution is the new algorithm-independent upper bounds for the observed posterior standard deviations (Lemmas 3–5) by bridging the information gain-based analysis in the noisy regime to the noise-free setting. Furthermore, as discussed in Section 3, these results have the potential to translate existing confidence bound-based algorithms for noisy settings into nearly-optimal noise-free variants beyond the analysis of GP-UCB.

**Related works.** Various existing works study the theory for the noisy GP bandits [Chowdhury and Gopalan, 2017, Li and Scarlett, 2022, Scarlett et al., 2017, Srinivas et al., 2010, Valko et al., 2013]. Regarding noise-free settings, to our knowledge, [Bull, 2011] is the first work that shows both the upper bound and the lower bound for simple regret via the expected improvement (EI) strategy. After that, the analysis of the cumulative regret is shown in [Lyu et al., 2019] with GP-UCB. Recently, Vakili [2022] conjectures the lower bound of the cumulative regret in the noise-free setting, suggesting a superior algorithm exists in the noise-free setting. Later, the following work [Li and Scarlett, 2024] formally validates the conjectured lower bound of [Vakili, 2022] [1]. Motivated by the lower bounds, several works [Flynn and Reeb, 2024, Kim and Sanz-Alonso, 2024, Iwazaki and Takeno, 2025a, Salgia et al., 2024] studied the improved algorithm to achieve superior regret to the result of [Lyu et al., 2019]. Although some of them propose the nearly-optimal algorithms [Iwazaki and Takeno, 2025a, Salgia et al., 2024], their algorithms are based on a non-adaptive sampling scheme, whose inferior performance has been reported in the existing work [Bogunovic et al., 2022, Iwazaki and Takeno, 2025b, Li and Scarlett, 2022]. Here, our motivation is to establish the nearly-optimal theory under the fully adaptive nature of GP-UCB; however, our proof relates to the analysis in [Iwazaki and Takeno, 2025a] for the non-adaptive maximum variance reduction algorithm. Their analysis includes the noise-free setting as a special case and provides nearly-optimal regret under broader varying noise variance settings. However, its applicability is limited to the maximum variance reduction algorithm. Our analysis can be interpreted as a refined version of that in [Iwazaki and Takeno, 2025a] by focusing on the noise-free setting. Finally, although our paper studies the frequentist assumption that the underlying function is fixed, our core results (Lemmas 3–5) are also applicable to the regret analysis in the Bayesian setting, whose underlying function is drawn from a known GP [De Freitas et al., 2012, Grünewälder et al., 2010, Russo and Van Roy, 2014a,b, Scarlett, 2018, Srinivas et al., 2010].

---

[1]Although Li and Scarlett [2024] considers the cascading structure of the observation process, the standard noise-free setting is the special case of their setting.

Table 1: Comparison between existing noise-free algorithms' guarantees for cumulative regret and our result (adapted from [Iwazaki and Takeno, 2025a]). As with the table in [Iwazaki and Takeno, 2025a], the smoothness parameter $\nu$ of the Matérn kernel and $\alpha > 0$ in PE are assumed to be $\nu > 1/2$ and an arbitrary fixed constant, respectively. Furthermore, all the parameters ($d$, $\ell$, $\nu$, and $B$) except for $T$ are assumed to be $\Theta(1)$. "Type" column shows that the regret guarantee is (D)eterministic or (P)robabilistic. Here, a regret bound is labeled "deterministic" if it holds for all possible realizations of the inputs, without relying on probabilistic assumptions. Here, $\widetilde{O}(\cdot)$ hides polylogarithmic factors in $T$.

| Algorithm | Regret (SE) | Regret (Matérn) | | | Type |
|---|---|---|---|---|---|
| | | $\nu < d$ | $\nu = d$ | $\nu > d$ | |
| GP-UCB [Lyu et al., 2019] [Kim and Sanz-Alonso, 2024] | $O\left(\sqrt{T \ln^d T}\right)$ | $\widetilde{O}\left(T^{\frac{\nu+d}{2\nu+d}}\right)$ | | | D |
| Explore-then-Commit [Vakili, 2022] | N/A | $\widetilde{O}\left(T^{\frac{d}{\nu+d}}\right)$ | | | P |
| Kernel-AMM-UCB [Flynn and Reeb, 2024] | $O\left(\ln^{d+1} T\right)$ | $\widetilde{O}\left(T^{\frac{\nu d+d^2}{2\nu^2+2\nu d+d^2}}\right)$ | | | D |
| REDS [Salgia et al., 2024] | N/A | $\widetilde{O}\left(T^{\frac{d-\nu}{d}}\right)$ | $O\left(\ln^{\frac{5}{2}} T\right)$ | $O\left(\ln^{\frac{3}{2}} T\right)$ | P |
| PE [Iwazaki and Takeno, 2025a] | $O(\ln T)$ | $\widetilde{O}\left(T^{\frac{d-\nu}{d}}\right)$ | $O\left(\ln^{2+\alpha} T\right)$ | $O(\ln T)$ | D |
| **GP-UCB (Our analysis)** | $O(1)$ | $\widetilde{O}\left(T^{\frac{d-\nu}{d}}\right)$ | $O\left(\ln^2 T\right)$ | $O(1)$ | D |
| Conjectured Lower Bound [Vakili, 2022] | N/A | $\Omega\left(T^{\frac{d-\nu}{d}}\right)$ | $\Omega(\ln T)$ | $\Omega(1)$ | N/A |
| Lower Bound [Li and Scarlett, 2024] | N/A | $\Omega\left(T^{\frac{d-\nu}{d}}\right)$ | $\Omega(1)$ | $\Omega(1)$ | N/A |

## 2  Preliminaries

**Noise-free GP bandit problem.**   We study the GP bandit problem under noise-free observations. Let $\mathcal{X} \subset \mathbb{R}^d$ be a compact input domain, and consider a black-box objective function $f : \mathcal{X} \to \mathbb{R}$ that can only be evaluated point-wise. At each step $t \in \mathbb{N}_+$, the learner selects a query point $\boldsymbol{x}_t \in \mathcal{X}$ and observes its function value $f(\boldsymbol{x}_t)$. After $T$ steps, the performance of the learner is measured using either the cumulative regret $R_T$ or the simple regret $r_T$, which are respectively defined as follows:

$$R_T = \sum_{t=1}^{T} f(\boldsymbol{x}^*) - f(\boldsymbol{x}_t), \tag{1}$$

$$r_T = f(\boldsymbol{x}^*) - f(\widehat{\boldsymbol{x}}_T). \tag{2}$$

Here, $\boldsymbol{x}^* \in \arg\max_{\boldsymbol{x} \in \mathcal{X}} f(\boldsymbol{x})$ is a global maximizer, and $\widehat{\boldsymbol{x}}_T$ denotes the estimated maximizer returned by the algorithm at the end of the final step $T$.

**Gaussian process model.**   To construct the GP-bandit algorithm, the GP model [Rasmussen and Williams, 2005] plays a central role in balancing the trade-off between exploration and exploitation. First, we adopt a Gaussian process prior with zero mean and covariance (kernel) function $k : \mathcal{X} \times \mathcal{X} \to \mathbb{R}$. Then, given a sequence of queried points $\mathbf{X}_t = (\boldsymbol{x}_1, \ldots, \boldsymbol{x}_t)$ and their corresponding evaluations $\boldsymbol{f}(\mathbf{X}_t) = (f(\boldsymbol{x}_1), \ldots, f(\boldsymbol{x}_t))$, the posterior mean $\mu(\boldsymbol{x}; \mathbf{X}_t)$ and variance $\sigma^2(\boldsymbol{x}; \mathbf{X}_t)$ of $f(\boldsymbol{x})$ at a new point $\boldsymbol{x} \in \mathcal{X}$ are

$$\mu(\boldsymbol{x}; \mathbf{X}_t) = \boldsymbol{k}(\boldsymbol{x}, \mathcal{E}(\mathbf{X}_t))^\top \mathbf{K}(\mathcal{E}(\mathbf{X}_t), \mathcal{E}(\mathbf{X}_t))^{-1} \boldsymbol{f}(\mathcal{E}(\mathbf{X}_t)), \tag{3}$$

$$\sigma^2(\boldsymbol{x}; \mathbf{X}_t) = k(\boldsymbol{x}, \boldsymbol{x}) - \boldsymbol{k}(\boldsymbol{x}, \mathcal{E}(\mathbf{X}_t))^\top \mathbf{K}(\mathcal{E}(\mathbf{X}_t), \mathcal{E}(\mathbf{X}_t))^{-1} \boldsymbol{k}(\boldsymbol{x}, \mathcal{E}(\mathbf{X}_t)), \tag{4}$$

where $\boldsymbol{k}(\boldsymbol{x}, \mathcal{E}(\mathbf{X}_t)) = [k(\widetilde{\boldsymbol{x}}, \boldsymbol{x})]_{\widetilde{\boldsymbol{x}} \in \mathcal{E}(\mathbf{X}_t)}$ is the kernel vector, $\mathbf{K}(\mathcal{E}(\mathbf{X}_t), \mathcal{E}(\mathbf{X}_t))$ is the Gram matrix, and $\boldsymbol{f}(\mathcal{E}(\mathbf{X}_t)) = [f(\widetilde{\boldsymbol{x}})]_{\widetilde{\boldsymbol{x}} \in \mathcal{E}(\mathbf{X}_t)}$. In the above definition, $\mathcal{E}(\mathbf{X}_t)$ denotes the subset of $\mathbf{X}_t$ obtained by removing any fully correlated inputs (i.e., with zero posterior variance) with previous ones. Namely, we define $\mathcal{E}(\mathbf{X}_t)$ inductively as $\mathcal{E}(\mathbf{X}_t) = \mathcal{E}(\mathbf{X}_{t-1}) \cup \{\boldsymbol{x}_t\}$ if $\sigma^2(\boldsymbol{x}_t; \mathbf{X}_{t-1}) > 0$; otherwise,

Table 2: Comparison between existing noiseless algorithms' guarantees for simple regret and our result (adapted from [Iwazaki and Takeno, 2025a]). In the regrets of GP-UCB+, EXPLOIT+, MVR, and GP-UCB, $\alpha > 0$ and $C > 0$ are arbitrary fixed constants and some positive constants, respectively.

| Algorithm | Regret (SE) | Regret (Matérn) | Type |
|---|---|---|---|
| GP-EI [Bull, 2011] | N/A | $\widetilde{O}\left(T^{-\frac{\min\{1,\nu\}}{d}}\right)$ | D |
| GP-EI with $\epsilon$-Greedy [Bull, 2011] | N/A | $\widetilde{O}\left(T^{-\frac{\nu}{d}}\right)$ | P |
| GP-UCB [Lyu et al., 2019] [Kim and Sanz-Alonso, 2024] | $O\left(\sqrt{\frac{\ln^d T}{T}}\right)$ | $\widetilde{O}\left(T^{-\frac{\nu}{2\nu+d}}\right)$ | D |
| Kernel-AMM-UCB [Flynn and Reeb, 2024] | $O\left(\frac{\ln^{d+1} T}{T}\right)$ | $\widetilde{O}\left(T^{-\frac{\nu d+2\nu^2}{2\nu^2+2\nu d+d^2}}\right)$ | D |
| GP-UCB+, EXPLOIT+ [Kim and Sanz-Alonso, 2024] | $O\left(\exp\left(-CT^{\frac{1}{d}-\alpha}\right)\right)$ | $O\left(T^{-\frac{\nu}{d}+\alpha}\right)$ | P |
| MVR [Iwazaki and Takeno, 2025a] | $O\left(\exp\left(-\frac{1}{2}T^{\frac{1}{d+1}}\ln^{-\alpha}T\right)\right)$ | $\widetilde{O}\left(T^{-\frac{\nu}{d}}\right)$ | D |
| **GP-UCB** **(Our analysis)** | $O\left(\sqrt{T}\exp\left(-\frac{1}{2}CT^{\frac{1}{d+1}}\right)\right)$ | $\widetilde{O}\left(T^{-\frac{\nu}{d}}\right)$ | D |
| Lower Bound [Bull, 2011] | N/A | $\Omega\left(T^{-\frac{\nu}{d}}\right)$ | N/A |

$\mathcal{E}(\mathbf{X}_t) = \mathcal{E}(\mathbf{X}_{t-1})$. Here, we define $\mathcal{E}(\mathbf{X}_1) = \mathbf{X}_1$. Note that if there are no duplications in the input sequence $\boldsymbol{x}_1, \ldots, \boldsymbol{x}_t$, then $\mathcal{E}(\mathbf{X}_t) = \mathbf{X}_t$ holds under commonly used kernel (covariance) functions, such as squared exponential and Matérn kernels (precisely defined in the next paragraph). Furthermore, for the ease of notation, we set $\mu(\boldsymbol{x}; \mathbf{X}) = 0$ and $\sigma^2(\boldsymbol{x}; \mathbf{X}) = k(\boldsymbol{x}, \boldsymbol{x})$ for $\mathbf{X} = \emptyset$.

**Kernel function and information gain.** Regarding the choice of the kernel function, we focus on the squared exponential (SE) kernel

$$k_{\text{SE}}(\boldsymbol{x}, \widetilde{\boldsymbol{x}}) = \exp\left(-\frac{\|\boldsymbol{x} - \widetilde{\boldsymbol{x}}\|_2^2}{2\ell^2}\right), \tag{5}$$

and the Matérn kernel

$$k_{\text{Matérn}}(\boldsymbol{x}, \widetilde{\boldsymbol{x}}) = \frac{2^{1-\nu}}{\Gamma(\nu)}\left(\frac{\sqrt{2\nu}\|\boldsymbol{x} - \widetilde{\boldsymbol{x}}\|_2}{\ell}\right)^\nu J_\nu\left(\frac{\sqrt{2\nu}\|\boldsymbol{x} - \widetilde{\boldsymbol{x}}\|_2}{\ell}\right), \tag{6}$$

where $\ell > 0$ and $\nu > 0$ are the lengthscale and smoothness parameter, respectively. Furthermore, $J_\nu$ and $\Gamma$ are the modified Bessel and Gamma functions, respectively. These two kernels are commonly used and analyzed in GP-bandits [Scarlett et al., 2017, Srinivas et al., 2010]. The convergence rate of the function estimation in GP regression depends on the choice of kernel, which in turn affects the resulting regret upper bound. Therefore, to capture the problem complexity in kernel-dependent manner, the following kernel-dependent information theoretic quantity $\gamma_T(\lambda^2)$ is often employed in the analysis of GP bandits:

$$\gamma_T(\lambda^2) = \sup_{\boldsymbol{x}_1, \ldots, \boldsymbol{x}_T \in \mathcal{X}} \frac{1}{2}\ln\det(\boldsymbol{I}_T + \lambda^{-2}\mathbf{K}(\mathbf{X}_T, \mathbf{X}_T)), \tag{7}$$

where $\lambda > 0$ and $\boldsymbol{I}_T$ are any positive parameter and $T \times T$-identity matrix, respectively. The quantity $\gamma_T(\lambda^2)$ is called *maximum information gain* (MIG) [Srinivas et al., 2010] since the quantity $\frac{1}{2}\ln\det(\boldsymbol{I}_T + \lambda^{-2}\mathbf{K}(\mathbf{X}_T, \mathbf{X}_T))$ represents the mutual information between the underlying function $f$ and training outputs under the noisy-GP model with variance parameter $\lambda^2$. The increasing speed of MIG is analyzed in several commonly used kernels. For example, $\gamma_T(\lambda^2) = O(\ln^{d+1}(T/\lambda^2))$ and $\gamma_T(\lambda^2) = \widetilde{O}((T/\lambda^2)^{\frac{d}{2\nu+d}})$ under $k = k_{\text{SE}}$ and $k = k_{\text{Matérn}}$ with $\nu > 1/2$, respectively [Vakili et al., 2021c][2].

---

[2]These orders hold as $T \to \infty$, $\lambda \to 0$.

---

**Algorithm 1** Gaussian process upper confidence bound (GP-UCB) for noise-free setting.

---

**Require:** Compact input domain $\mathcal{X} \subset \mathbb{R}^d$, Kernel function $k$, and RKHS norm upper bound $B \in (0, \infty)$.
1: $\mathbf{X}_0 \leftarrow \emptyset, \beta^{1/2} \leftarrow B$.
2: **for** $t = 1, 2, \ldots$ **do**
3:    $x_t \leftarrow \arg\max_{x \in \mathcal{X}} \mu(x; \mathbf{X}_{t-1}) + \beta^{1/2} \sigma(x; \mathbf{X}_{t-1})$.
4:    Observe $f(x_t)$ and update the posterior mean and variance.
5: **end for**

---

**Regularity assumption.** It is hopeless to derive meaningful guarantees without further assumptions about $f$. To obtain a valid estimate of $f$ through the GP-model, we assume that $f$ lies in the known reproducing kernel Hilbert space (RKHS) [Aronszajn, 1950] corresponding to the kernel $k$ and has bounded norm. The formal description is given below.

**Assumption 1.** *The objective function $f$ lies in the reproducing kernel Hilbert space (RKHS) associated with a known positive definite kernel $k : \mathcal{X} \times \mathcal{X} \to \mathbb{R}$. We assume that $k(x, x) \leq 1$ for all $x \in \mathcal{X}$, and that the RKHS norm $\|f\|_k$ satisfies $\|f\|_k \leq B < \infty$.*

This is a standard assumption in the GP-bandit literature [Chowdhury and Gopalan, 2017, Scarlett et al., 2017, Srinivas et al., 2010], and is leveraged to derive the confidence bound of $f$ [Kanagawa et al., 2018, Lyu et al., 2019, Vakili et al., 2021a]. Here, RKHS associated with kernel $k$ is given as the closure of the linear span: $\mathcal{H}_k^{(\text{pre})} := \{\sum_{i=1}^n c_i k(x^{(i)}, \cdot) \mid n \in \mathbb{N}_+, c_1, \ldots, c_n \in \mathbb{R}, x^{(1)}, \ldots, x^{(n)} \in \mathcal{X}\}$ [Kanagawa et al., 2018]. Therefore, intuitively, under Assumption 1, we can interpret that the basic properties of the objective function $f$, such as continuity and smoothness, are encoded through the choice of the kernel function $k$.

**Gaussian process upper confidence bound.** GP-UCB [Srinivas et al., 2010] is a widely used algorithm for the noisy GP bandit setting. Its noiseless variant was proposed by Lyu et al. [2019], and is outlined in Algorithm 1. Their analysis largely follows the framework developed for the noisy case in [Srinivas et al., 2010], but differs in the confidence bound, which is strictly tighter than that in the noisy setting. Although this refinement of the confidence bound leads to superior regret compared with the noisy setting, the resulting regret is still strictly sub-optimal in the noise-free setting. This fact suggests that we require the other fundamental modification from the proof in [Srinivas et al., 2010].

## 3 Refined Regret Upper Bound for Noise-Free GP-UCB

The following theorem describes our main results, which show the nearly-optimal regret upper bound for GP-UCB.

**Theorem 1** (Refined cumulative regret upper bound for GP-UCB)**.** *Fix any compact input domain $\mathcal{X} \subset \mathbb{R}^d$. Suppose $B$, $d$, $\ell$, and $\nu$ are fixed constants. Then, when running Algorithm 1 under Assumption 1, the following two statements hold for any $T \in \mathbb{N}_+$:*

- *If $k = k_{\text{SE}}$, the regret $R_T$ satisfies $R_T = O(1)$.*

- *If $k = k_{\text{Matérn}}$ with $\nu > 1/2$, the regret $R_T$ satisfies*

$$R_T = \begin{cases} \widetilde{O}\left(T^{\frac{d-\nu}{d}}\right) & \text{if } d > \nu, \\ O(\ln^2 T) & \text{if } d = \nu, \\ O(1) & \text{if } d < \nu. \end{cases} \tag{8}$$

*The implied constants may depend on $B$, $d$, $\ell$, $\nu$, and the diameter of $\mathcal{X}$.*

**Theorem 2** (Refined simple regret upper bound for GP-UCB)**.** *Fix any compact input domain $\mathcal{X} \subset \mathbb{R}^d$. Suppose $B$, $L$, $d$, $\ell$, and $\nu$ are fixed constants. Then, when running Algorithm 1 under Assumption 1, the following two statements hold by setting the estimated maximizer $\widehat{x}_T$ as $\widehat{x}_T \in \arg\max_{x \in \{x_1, \ldots, x_T\}} f(x)$:*

- *If $k = k_{\text{SE}}$, $r_T = O\left(\sqrt{T}\exp\left(-\frac{1}{2}CT^{\frac{1}{d+1}}\right)\right)$.*

- *If $k = k_{\text{Matérn}}$ with $\nu > 1/2$, $r_T = \widetilde{O}\left(T^{-\frac{\nu}{d}}\right)$.*

*The implied constants depend on $B$, $d$, $\ell$, $\nu$, and the diameter of $X$. Furthermore, the constant $C > 0$ depends on $d$, $\ell$, $\nu$, and the diameter of $X$[3][4].*

**Proof sketch.** Our key technical results are the new analysis of the cumulative posterior standard deviation $\sum_{t=1}^{T} \sigma(\boldsymbol{x}_t; \mathbf{X}_{t-1})$ and its minimum $\min_{t \in [T]} \sigma(\boldsymbol{x}_t; \mathbf{X}_{t-1})$, which plays an important role in the theoretical analysis of GP bandits. Indeed, following the standard analysis of GP-UCB, we have the following upper bounds of regrets by combining the UCB-selection rule with the existing noise-free confidence bound (e.g., Lemma 11 in [Lyu et al., 2019] or Proposition 1 in [Vakili et al., 2021a]):

$$R_T = \sum_{t=1}^{T} f(\boldsymbol{x}^*) - f(\boldsymbol{x}_t) \leq 2B \sum_{t=1}^{T} \sigma(\boldsymbol{x}_t; \mathbf{X}_{t-1}), \tag{9}$$

$$r_T = \min_{t \in [T]} f(\boldsymbol{x}^*) - f(\boldsymbol{x}_t) \leq 2B \min_{t \in [T]} \sigma(\boldsymbol{x}_t; \mathbf{X}_{t-1}). \tag{10}$$

From the above inequalities, we observe that the tighter upper bounds of $\sum_{t=1}^{T} \sigma(\boldsymbol{x}_t; \mathbf{X}_{t-1})$ and $\min_{t \in [T]} \sigma(\boldsymbol{x}_t; \mathbf{X}_{t-1})$ directly yield the tighter regret upper bounds of GP-UCB. Lemma 3 below is our main technical contribution, which gives the refined upper bounds of $\sum_{t=1}^{T} \sigma(\boldsymbol{x}_t; \mathbf{X}_{t-1})$ and $\min_{t \in [T]} \sigma(\boldsymbol{x}_t; \mathbf{X}_{t-1})$.

**Lemma 3** (Posterior standard deviation upper bound for SE and Matérn kernel). *Fix any compact input domain $X \subset \mathbb{R}^d$, and kernel function $k : X \times X \to \mathbb{R}$ that satisfies $k(\boldsymbol{x}, \boldsymbol{x}) \leq 1$ for all $\boldsymbol{x} \in X$. Then, the following statements hold for any $T \in \mathbb{N}_+$ and any input sequence $\boldsymbol{x}_1, \ldots, \boldsymbol{x}_T \in X$:*

- *For $k = k_{\text{SE}}$, we have*

$$\min_{t \in [T]} \sigma(\boldsymbol{x}_t; \mathbf{X}_{t-1}) = O\left(\sqrt{T}\exp\left(-\frac{1}{2}CT^{\frac{1}{d+1}}\right)\right) \text{ and } \sum_{t=1}^{T} \sigma(\boldsymbol{x}_t; \mathbf{X}_{t-1}) = O(1). \tag{11}$$

- *For $k = k_{\text{Matérn}}$ with $\nu > 1/2$, we have*

$$\min_{t \in [T]} \sigma(\boldsymbol{x}_t; \mathbf{X}_{t-1}) = O\left(T^{-\frac{\nu}{d}} \ln^{\frac{\nu}{d}} T\right), \tag{12}$$

$$\sum_{t=1}^{T} \sigma(\boldsymbol{x}_t; \mathbf{X}_{t-1}) = \begin{cases} O\left(T^{\frac{d-\nu}{d}} \ln^{\frac{\nu}{d}} T\right) & \text{if } d > \nu, \\ O\left(\ln^2 T\right) & \text{if } d = \nu, \\ O(1) & \text{if } d < \nu. \end{cases} \tag{13}$$

*The constant $C > 0$ and the implied constants depend on $d$, $\ell$, $\nu$, and the diameter of $X$.*

The full proof of Lemma 3 is given in Appendix A.2. We will also provide the proof sketch in the next section. Combining the above equations with Eqs. (9) and (10), we obtain the statements in Theorems 1 and 2.

---

[3]As described in Section 3.1, the constant $C$ arises from the implied constants in the upper bound of MIG [Vakili et al., 2021c], which depends on $d$, $\ell$, $\nu$, and the diameter of $X$

[4]Our results (Theorems 1 and 2) heavily rely on the upper bound of the MIG provided in [Vakili et al., 2021c], which relies on the uniform boundness assumption of the eigenfunctions of the kernel. The validity of the uniform boundness assumption is doubted by Janz [2022] under a general compact input domain $X$. Although our results are based on the upper bound of MIG in [Vakili et al., 2021c], our proof strategy is also applicable for deriving nearly-optimal regrets based on the recent analysis of MIG [Iwazaki, 2025] without the uniform boundness assumption. Specifically, then, the orders of the resulting regrets are the same as those in Theorems 1 and 2 for $k_{\text{SE}}$ and $k_{\text{Matérn}}$ with $d < \nu$. As for the case under $k = k_{\text{Matérn}}$ with $d \geq \nu$, we can also obtain $R_T = \widetilde{O}(T^{(d-\nu)/d})$ and $r_T = \widetilde{O}(T^{-\nu/d})$, while they suffer from additional polylogarithmic terms.

**Relation to the existing research in noise-free setting.** Lemma 3 resolves the open problem raised by the existing noise-free setting literature [Li and Scarlett, 2024, Vakili, 2022]. First, Vakili [2022] conjectured that the quantity $\sum_{t=1}^{T} \sigma(x_t; \mathbf{X}_{t-1})$ under $k = k_{\text{Matérn}}$ can attain the following upper bound:

$$\sum_{t=1}^{T} \sigma(x_t; \mathbf{X}_{t-1}) = \begin{cases} O\left(T^{\frac{d-\nu}{d}}\right) & \text{if } d > \nu, \\ O\left(\ln T\right) & \text{if } d = \nu, \\ O(1) & \text{if } d < \nu. \end{cases} \tag{14}$$

Although the above conjecture is partially validated under a specific non-adaptive algorithm [Li and Scarlett, 2024, Iwazaki and Takeno, 2025a, Salgia et al., 2024], the correctness of this conjecture under a general algorithm has been an open problem [Li and Scarlett, 2024]. Lemma 3 answers this open problem with Eq. (13), which matches conjectured upper bound up to polylogarithmic factors.

**Generality of Lemma 3.** We would like to highlight that Lemma 3 always holds for any input sequence, in contrast to the existing algorithm-specific upper bounds [Iwazaki and Takeno, 2025a, Salgia et al., 2024]. Since the existing noisy GP bandits theory often leverage the upper bound of $\min_{t \in [T]} \sigma(x_t; \mathbf{X}_{t-1})$ or $\sum_{t=1}^{T} \sigma(x_t; \mathbf{X}_{t-1})$ from [Srinivas et al., 2010], we expect that many existing theoretical results in the noisy setting can be extended to the corresponding noise-free setting by directly replacing the existing noisy upper bounds of [Srinivas et al., 2010] with Lemma 3. For example, the analysis for GP-Thompson sampling (GP-TS) [Chowdhury and Gopalan, 2017], GP-UCB and GP-TS under Bayesian setting [Russo and Van Roy, 2014a, Srinivas et al., 2010], contextual setting [Krause and Ong, 2011], GP-based level-set estimation [Gotovos et al., 2013], multi-objective setting [Zuluaga et al., 2016], robust formulation [Bogunovic et al., 2018], and so on.

**Extension to other kernel functions.** Lemma 3 is limited to the SE and Matérn kernels. However, our proof strategy in Section 3.1 can be applied to any kernel function if we know the joint dependence of $T$ and the noise-variance parameter $\lambda^2$ in MIG. For example, we can apply our proof strategy for the neural tangent kernel (NTK) [Jacot et al., 2018] by leveraging the existing upper bound of MIG under NTK [Iwazaki and Suzumura, 2024, Kassraie and Krause, 2022, Kassraie et al., 2022, Vakili et al., 2021b]. We expect that such an extension will benefit the theoretical guarantees of neural network-based bandit algorithms under a noise-free setting and will be one of the interesting research directions.

### 3.1 Proof Sketch of Lemma 3

In this subsection, we describe the proof sketch of Lemma 3, while we give its full proof in Appendix A.2. Below, to prove Lemma 3, we consider the more general lemmas, which bridge the MIG to noise-free posterior standard deviations.

**Lemma 4** (General upper bound for the minimum posterior standard deviation). *Fix any input domain $\mathcal{X}$ and any $\overline{T} \geq 2$. Let $(\lambda_t)_{t \geq \overline{T}}$ be a strictly positive sequence such that $\gamma_t(\lambda_t^2) \leq (t-1)/3$ for all $t \geq \overline{T}$. Then, $\min_{t \in [T]} \sigma(x_t; \mathbf{X}_{t-1}) \leq \lambda_T$ holds for any $T \geq \overline{T}$ and any sequence $x_1, \ldots, x_T \in \mathcal{X}$.*

**Lemma 5** (General upper bound for the cumulative posterior standard deviations). *Fix any input domain $\mathcal{X}$, any $\overline{T} \geq 2$, and any kernel function $k : \mathcal{X} \times \mathcal{X} \to \mathbb{R}$ that satisfies $k(x, x) \leq 1$ for all $x \in \mathcal{X}$. Let $(\lambda_t)_{t \geq \overline{T}}$ be a strictly positive sequence such that $\gamma_t(\lambda_t^2) \leq (t-1)/3$ for all $t \geq \overline{T}$. Then, the following inequality holds for any $T \in \mathbb{N}_+$ and any sequence $x_1, \ldots, x_T \in \mathcal{X}$:*

$$\sum_{t=1}^{T} \sigma(x_t; \mathbf{X}_{t-1}) \leq \overline{T} - 1 + \sum_{t=\overline{T}}^{T} \lambda_t. \tag{15}$$

These lemmas hold for any kernel k satisfying $k(x, x) \leq 1$, which includes most standard kernels after normalization. Roughly speaking, the above lemmas suggest that $\min_{t \in [T]} \sigma(x_t; \mathbf{X}_{t-1}) \lesssim \lambda_T$ and $\sum_{t=1}^{T} \sigma(x_t; \mathbf{X}_{t-1}) \lesssim \sum_{t=1}^{T} \lambda_t$ holds as far as the corresponding MIG $\gamma_t(\lambda_t^2)$ does not increase super-linearly. Note that the MIG $\gamma_t(\lambda_t^2)$ monotonically increases as $\lambda_t^2$ decreases, which implies that the tightest upper bound is obtained by setting $\lambda_t^2$ as $\gamma_t(\lambda_t^2) = (t-1)/3$. By relying on the existing upper bound of MIG [Vakili et al., 2021c], we can confirm that the condition $\forall t \geq \overline{T}, \gamma_t(\lambda_t^2) \leq (t-1)/3$

of the above lemmas holds with $\lambda_t^2 = O(t \exp(-Ct^{\frac{1}{d+1}}))$ and $\lambda_t^2 = O(t^{-\frac{2\nu}{d}}(\ln t)^{\frac{2\nu}{d}})$ for $k = k_{\mathrm{SE}}$ and $k = k_{\mathrm{Matérn}}$, respectively. Here, the constants $C$ and $\overline{T}$ are determined based on the implied constant of the upper bound of MIG. See Appendix A.2 for details. Lemma 3 follows from the aforementioned setting of $\lambda_t^2$, and Lemmas 4 and 5. Below, we give the proofs for Lemmas 4 and 5.

**Proof of Lemma 4.** Instead of directly treating the noise-free posterior standard deviation, we study its upper bound with the posterior standard deviation of some noisy GP model. Here, let us denote $\sigma_{\lambda^2}^2(\boldsymbol{x}; \mathbf{X}_{t-1})$ as the posterior variance under the noisy GP-model with the strictly positive variance parameter $\lambda^2 > 0$, which is defined as

$$\sigma_{\lambda^2}^2(\boldsymbol{x}; \mathbf{X}_{t-1}) = k(\boldsymbol{x}, \boldsymbol{x}) - \boldsymbol{k}(\boldsymbol{x}, \mathbf{X}_{t-1})^\top [\mathbf{K}(\mathbf{X}_{t-1}, \mathbf{X}_{t-1}) + \lambda^2 I_{t-1}]^{-1} \boldsymbol{k}(\boldsymbol{x}, \mathbf{X}_{t-1}). \qquad (16)$$

Since the posterior variance is monotonic for the variance parameter, we have $\sigma^2(\boldsymbol{x}_t; \mathbf{X}_{t-1}) \leq \sigma_{\lambda_T^2}^2(\boldsymbol{x}_t; \mathbf{X}_{t-1})$ for all $t \in [T]$. Next, we obtain the upper bound of $\sigma_{\lambda_T^2}^2(\boldsymbol{x}; \mathbf{X}_{t-1})$ based on the following lemma, which is the main component of the proof of Lemmas 4 and 5.

**Lemma 6** (Elliptical potential count lemma, Lemma D.9 in [Flynn and Reeb, 2024] or Lemma 3.3 in [Iwazaki and Takeno, 2025a]). *Fix any $T \in \mathbb{N}_+$, any sequence $\boldsymbol{x}_1, \ldots, \boldsymbol{x}_T \in \mathcal{X}$, and $\lambda > 0$. Define $\mathcal{T}$ as $\mathcal{T} = \{t \in [T] \mid \lambda^{-1}\sigma_{\lambda^2}(\boldsymbol{x}_t; \mathbf{X}_{t-1}) > 1\}$, where $\mathbf{X}_{t-1} = (\boldsymbol{x}_1, \ldots, \boldsymbol{x}_{t-1})$. Then, the number of elements of $\mathcal{T}$ satisfies $|\mathcal{T}| \leq 3\gamma_T(\lambda^2)$.*

The above lemma implies that the set $\mathcal{T}^c := \{t \in [T] \mid \sigma_{\lambda_T^2}(\boldsymbol{x}_t; \mathbf{X}_{t-1}) \leq \lambda_T\}$ satisfies $|\mathcal{T}^c| = |[T] \setminus \mathcal{T}| \geq T - 3\gamma_T(\lambda_T^2)$. Therefore, for any $T \geq \overline{T}$, $|\mathcal{T}^c| \geq 1$ holds from the condition $\gamma_T(\lambda_T^2) \leq (T-1)/3$. This implies there exists some $\widetilde{t} \in [T]$ such that $\sigma(\boldsymbol{x}_{\widetilde{t}}; \mathbf{X}_{\widetilde{t}-1}) \leq \sigma_{\lambda_T^2}(\boldsymbol{x}_{\widetilde{t}}; \mathbf{X}_{\widetilde{t}-1}) \leq \lambda_T$; therefore, $\min_{t \in [T]} \sigma(\boldsymbol{x}_t; \mathbf{X}_{t-1}) \leq \sigma(\boldsymbol{x}_{\widetilde{t}}; \mathbf{X}_{\widetilde{t}-1}) \leq \lambda_T$ holds for all $T \geq \overline{T}$. $\qquad \square$

**Proof of Lemma 5.** Overall, the proof strategy of this lemma is to repeatedly apply the proof of Lemma 4 by leveraging the monotonicity of the posterior variance against training inputs. First, if $T < \overline{T}$, Eq. (15) is clearly holds from the assumption $\forall \boldsymbol{x} \in \mathcal{X}, k(\boldsymbol{x}, \boldsymbol{x}) \leq 1$. Hereafter, we focus on $T \geq \overline{T}$. By following the same argument of Lemma 4, we can confirm that there exists the index $\widetilde{t}_T \leq T$ such that $\sigma(\boldsymbol{x}_{\widetilde{t}_T}; \mathbf{X}_{\widetilde{t}_T-1}) \leq \sigma_{\lambda_T^2}(\boldsymbol{x}_{\widetilde{t}_T}; \mathbf{X}_{\widetilde{t}_T-1}) \leq \lambda_T$. Here, we define the new sequence $(\boldsymbol{x}_t^{(T-1)})_{t \in [T-1]}$ as the sequence that $\boldsymbol{x}_{\widetilde{t}}$ is eliminated from $(\boldsymbol{x}_t)_{t \in [T]}$; namely, we set $\boldsymbol{x}_t^{(T-1)} = \mathbb{1}\{t < \widetilde{t}_T\}\boldsymbol{x}_t + \mathbb{1}\{t \geq \widetilde{t}_T\}\boldsymbol{x}_{t+1}$ for any $t \in [T-1]$. Furthermore, we define $\mathbf{X}_t^{(T-1)} = (\boldsymbol{x}_1^{(T-1)}, \ldots, \boldsymbol{x}_t^{(T-1)})$. From this construction of $\mathbf{X}_t^{(T-1)}$, we can observe the following two facts:

- For any $t < \widetilde{t}_T$, we have $\sigma(\boldsymbol{x}_t; \mathbf{X}_{t-1}) = \sigma\left(\boldsymbol{x}_t^{(T-1)}; \mathbf{X}_{t-1}^{(T-1)}\right)$, since $\boldsymbol{x}_t^{(T-1)} = \boldsymbol{x}_t$ and $\mathbf{X}_{t-1}^{(T-1)} = \mathbf{X}_{t-1}$.

- For any $t > \widetilde{t}_T$, we have $\sigma(\boldsymbol{x}_t; \mathbf{X}_{t-1}) \leq \sigma\left(\boldsymbol{x}_{t-1}^{(T-1)}; \mathbf{X}_{t-2}^{(T-1)}\right)$, since $\boldsymbol{x}_{t-1}^{(T-1)} = \boldsymbol{x}_t$ and $\mathbf{X}_{t-2}^{(T-1)} \subset \mathbf{X}_{t-1}$ from the definition of $\mathbf{X}_t^{(T-1)}$.

From the above two facts, we have

$$\sum_{t=1}^T \sigma(\boldsymbol{x}_t; \mathbf{X}_{t-1}) \leq \sum_{t \in [T] \setminus \{\widetilde{t}_T\}} \sigma(\boldsymbol{x}_t; \mathbf{X}_{t-1}) + \lambda_T \leq \sum_{t \in [T-1]} \sigma\left(\boldsymbol{x}_t^{(T-1)}; \mathbf{X}_{t-1}^{(T-1)}\right) + \lambda_T. \qquad (17)$$

Then, we observe that there exists the index $\widetilde{t}_{T-1} \leq T-1$ such that $\sigma(\boldsymbol{x}_{\widetilde{t}_{T-1}}^{(T-1)}; \mathbf{X}_{\widetilde{t}_{T-1}-1}^{(T-1)}) \leq \sigma_{\lambda_{T-1}^2}(\boldsymbol{x}_{\widetilde{t}_{T-1}}^{(T-1)}; \mathbf{X}_{\widetilde{t}_{T-1}-1}^{(T-1)}) \leq \lambda_{T-1}$ by the application of Lemma 6 for the new sequence $(\boldsymbol{x}_t^{(T-1)})$. Again, by setting $\boldsymbol{x}_t^{(T-2)} = \mathbb{1}\{t < \widetilde{t}_{T-1}\}\boldsymbol{x}_t^{(T-1)} + \mathbb{1}\{t \geq \widetilde{t}_{T-1}\}\boldsymbol{x}_{t+1}^{(T-1)}$ and $\mathbf{X}_t^{(T-2)} = (\boldsymbol{x}_1^{(T-2)}, \ldots, \boldsymbol{x}_t^{(T-2)})$ for any $t \in [T-2]$, we have

$$\sum_{t \in [T-1]} \sigma\left(\boldsymbol{x}_t^{(T-1)}; \mathbf{X}_{t-1}^{(T-1)}\right) \leq \sum_{t \in [T-1] \setminus \{\widetilde{t}_{T-1}\}} \sigma\left(\boldsymbol{x}_t^{(T-1)}; \mathbf{X}_{t-1}^{(T-1)}\right) + \lambda_{T-1} \qquad (18)$$

$$\leq \sum_{t \in [T-2]} \sigma\left(\boldsymbol{x}_t^{(T-2)}; \mathbf{X}_{t-1}^{(T-2)}\right) + \lambda_{T-1}. \qquad (19)$$

We can repeat the above arguments until we reach $\overline{T} - 1$. Then, the resulting upper bound becomes

$$\sum_{t=1}^{T} \sigma(\boldsymbol{x}_t; \mathbf{X}_{t-1}) \leq \sum_{t=1}^{\overline{T}-1} \sigma\left(\boldsymbol{x}_t^{(\overline{T}-1)}; \mathbf{X}_{t-1}^{(\overline{T}-1)}\right) + \sum_{t=\overline{T}}^{T} \lambda_t \leq \overline{T} - 1 + \sum_{t=\overline{T}}^{T} \lambda_t, \tag{20}$$

where the last inequality follows from $\sigma\left(\boldsymbol{x}_t^{(\overline{T}-1)}; \mathbf{X}_{t-1}^{(\overline{T}-1)}\right) \leq k(\boldsymbol{x}_t^{(\overline{T}-1)}, \boldsymbol{x}_t^{(\overline{T}-1)}) \leq 1$. $\qquad\square$

## 4 Discussion

Below, we discuss the remaining open questions in the noise-free setting.

- **Lower bound for squared exponential kernel.** While our results establish near-optimality for the Matérn kernel, the optimal simple regret rate under the SE kernel remains unknown. Since the existing upper bound $O(\ln^{d+1} T)$ for MIG [Vakili et al., 2021c] does not matches $O(\ln^{d/2} T)$ lower bound[5], we conjecture that further room for improvement also exists in the noise-free setting. Specifically, the exponent $d + 1$ of the MIG is reflected in the denominator of the exponential factors in our regret Eq. (2). Therefore, we conjecture that $O(\sqrt{T}\exp(-CT^{2/d}))$ regret is the best guarantee for the simple regret in the SE kernel.

- **Constant cumulative regret in Bayesian setting.** By using Lemma 3, we can prove the same cumulative regret as Eq. (1) up to a logarithmic factor in noise-free GP-UCB or GP-TS in the Bayesian setting [Srinivas et al., 2010, Russo and Van Roy, 2014a]. However, the confidence width parameter in the Bayesian setting must scale as $\beta^{1/2} = O(\sqrt{\ln T})$ to construct a valid confidence bound. This leads to $O(\sqrt{\ln T})$ regret in SE and Matérn kernel with $d < \nu$ under the Bayesian setting, whereas the frequentist counterpart guarantees constant $O(1)$ regret (Theorem 1). This is counterintuitive, since, as shown in existing analyses for noisy settings [Scarlett, 2018, Srinivas et al., 2010], Bayesian regret often achieves smaller values than the worst-case regret in the frequentist setting. An interesting direction for future work is to either design a Bayesian algorithm with constant regret or prove an $\Omega(\sqrt{\ln T})$ lower bound in the Bayesian setting.

- **GP-UCB in simple regret minimization.** Our analysis shows that GP-UCB achieves nearly optimal simple and cumulative regrets under the Matérn kernel. On the other hand, several algorithms have been proposed that focus on minimizing the simple regret. One of the most well-known examples is EI, which greedily minimizes the simple regret under a Bayesian modeling assumption of GP, and has demonstrated good empirical performance in various applications [Brochu et al., 2010, Snoek et al., 2012]. Based on our theoretical results, there exists no algorithm that can achieve strictly better simple regret than that of GP-UCB in the worst-case sense. Nevertheless, as far as we are aware, GP-UCB tends to exhibit inferior empirical performance compared to EI in simple regret minimization. See, Appendix B.2. It remains unclear whether this phenomena arises from constant factors or additional logarithmic terms in the theoretical regret upper bounds, or whether it reflects a more fundamental gap between worst-case analysis and empirical behavior.

## 5 Conclusion

This paper shows that GP-UCB achieves nearly-optimal regret by proving a new regret upper bound for noise-free GP bandits. The key theoretical component of our analysis is a tight upper bound on the posterior standard deviations of GP tailored to a noise-free setting (Lemma 3). As remarked in Section 3, Lemma 3 can be applicable beyond the analysis of GP-UCB. Specifically, we expect that many existing theoretical results for noisy GP bandit settings can be translated to the noise-free setting by replacing the existing noisy upper bound of the posterior standard deviations with Lemma 3. For this reason, we believe that our result marks an important step toward advancing the theory for noise-free GP bandit algorithms.

---

[5]This is derived from the best-known cumulative regret upper bound $R_T = \widetilde{O}(\sqrt{T\gamma_T(\lambda^2)})$ [Li and Scarlett, 2022, Salgia et al., 2021, Valko et al., 2013], and lower bound $R_T = \Omega(\sqrt{T \ln^{d/2} T})$ in the noisy setting [Scarlett et al., 2017].

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

# A Proofs for Section 3

## A.1 Proof of Theorems 1 and 2

We first formally describe the existing noise-free confidence bound.

**Lemma 7** (Deterministic confidence bound for noise-free setting, e.g., Corollary 3.11 in [Kanagawa et al., 2018], Lemma 11 in [Lyu et al., 2019], or Proposition 1 in [Vakili et al., 2021a]). *Suppose Assumption 1 holds. Then, for any sequence $(\boldsymbol{x}_t)_{t \in \mathbb{N}_+}$ on $\mathcal{X}$, the following statement holds:*

$$\forall t \in \mathbb{N}_+, \forall \boldsymbol{x} \in \mathcal{X}, |f(\boldsymbol{x}) - \mu(\boldsymbol{x}; \mathbf{X}_t)| \leq B\sigma(\boldsymbol{x}; \mathbf{X}_t), \tag{21}$$

*where $\mathbf{X}_t = (\boldsymbol{x}_1, \ldots, \boldsymbol{x}_t)$.*

Although the remaining parts of the proofs are well-known results of GP-UCB, we provide the details for completeness. Based on the above lemma, we show Eqs. (9) and (10). Regarding $R_T$, we have

$$R_T = \sum_{t=1}^{T} f(\boldsymbol{x}^*) - f(\boldsymbol{x}_t) \tag{22}$$

$$\leq \sum_{t=1}^{T} [\mu(\boldsymbol{x}^*; \mathbf{X}_t) + B\sigma(\boldsymbol{x}^*; \mathbf{X}_t)] - [\mu(\boldsymbol{x}_t; \mathbf{X}_t) - B\sigma(\boldsymbol{x}_t; \mathbf{X}_t)] \tag{23}$$

$$\leq \sum_{t=1}^{T} [\mu(\boldsymbol{x}_t; \mathbf{X}_t) + B\sigma(\boldsymbol{x}_t; \mathbf{X}_t)] - [\mu(\boldsymbol{x}_t; \mathbf{X}_t) - B\sigma(\boldsymbol{x}_t; \mathbf{X}_t)] \tag{24}$$

$$= 2B \sum_{t=1}^{T} \sigma(\boldsymbol{x}_t; \mathbf{X}_t), \tag{25}$$

where the first inequality follows from Lemma 7, and the second inequality follows from the UCB-selection rule for $\boldsymbol{x}_t$. Similarly to the case of cumulative regret, we have

$$r_T = f(\boldsymbol{x}^*) - f(\widehat{\boldsymbol{x}}_T) \tag{26}$$

$$\leq \min_{t \in [T]} f(\boldsymbol{x}^*) - f(\boldsymbol{x}_t) \tag{27}$$

$$\leq \min_{t \in [T]} [\mu(\boldsymbol{x}^*; \mathbf{X}_t) + B\sigma(\boldsymbol{x}^*; \mathbf{X}_t)] - [\mu(\boldsymbol{x}_t; \mathbf{X}_t) - B\sigma(\boldsymbol{x}_t; \mathbf{X}_t)] \tag{28}$$

$$\leq \min_{t \in [T]} [\mu(\boldsymbol{x}_t; \mathbf{X}_t) + B\sigma(\boldsymbol{x}_t; \mathbf{X}_t)] - [\mu(\boldsymbol{x}_t; \mathbf{X}_t) - B\sigma(\boldsymbol{x}_t; \mathbf{X}_t)] \tag{29}$$

$$= 2B \min_{t \in [T]} \sigma(\boldsymbol{x}_t; \mathbf{X}_t), \tag{30}$$

where the first inequality follows from the definition of $\widehat{\boldsymbol{x}}_T$. Finally, the desired results are obtained by combining the above inequalities with Eqs. (11)–(13). $\qquad\square$

## A.2 Proof of Lemma 3

We prove the following Lemma 8, which is a detailed version of Lemma 3 including the dependence against constant factors.

**Lemma 8** (Detailed version of posterior standard deviation upper bound for SE and Matérn kernel). *Fix any compact input domain $\mathcal{X} \subset \mathbb{R}^d$, and kernel function $k : \mathcal{X} \times \mathcal{X} \to \mathbb{R}$ that satisfies $k(\boldsymbol{x}, \boldsymbol{x}) \leq 1$ for all $\boldsymbol{x} \in \mathcal{X}$. Furthermore, let $C_{\mathrm{SE}}, C_{\mathrm{Mat}}, \underline{\lambda}_{\mathrm{SE}}, \underline{\lambda}_{\mathrm{Mat}} > 0, \underline{T}_{\mathrm{SE}}, \underline{T}_{\mathrm{Mat}} \geq 2$ be the constants[6] that satisfies $\forall \lambda \in (0, \underline{\lambda}_{\mathrm{SE}}], \forall t \geq \underline{T}_{\mathrm{SE}}, \gamma_t(\lambda^2) \leq C_{\mathrm{SE}}(\ln(t/\lambda^2))^{d+1}$ and $\forall \lambda \in (0, \underline{\lambda}_{\mathrm{Mat}}], \forall t \geq \underline{T}_{\mathrm{Mat}}, \gamma_t(\lambda^2) \leq C_{\mathrm{Mat}}(t/\lambda^2)^{\frac{d}{2\nu+d}}(\ln(t/\lambda^2))^{\frac{2\nu}{2\nu+d}}$ for $k = k_{\mathrm{SE}}$ and $k = k_{\mathrm{Matérn}}$, respectively. Then, the following statements hold for any $T \in \mathbb{N}_+$ and any input sequence $\boldsymbol{x}_1, \ldots, \boldsymbol{x}_T \in \mathcal{X}$:*

---

[6]The existence of these constants are guaranteed by the upper bound of MIG [Vakili et al., 2021c], which shows $\gamma_T(\lambda^2) = O(\ln^{d+1}(T/\lambda^2))$ and $\gamma_T(\lambda^2) = O((T/\lambda^2)^{\frac{d}{2\nu+d}} \ln^{\frac{2\nu}{2\nu+d}}(T/\lambda^2))$ (as $T \to \infty, \lambda \to 0$) under $k = k_{\mathrm{SE}}$ and $k = k_{\mathrm{Matérn}}$, respectively. Note that these constants do not depend on $T$, but may depend on $d, \ell, \nu$, and the diameter of $\mathcal{X}$.

- *For $k = k_{\text{SE}}$,*

$$\min_{t \in [T]} \sigma(\boldsymbol{x}_t; \mathbf{X}_{t-1}) \leq \begin{cases} 1 & \text{if } T < \overline{T}_{\text{SE}}, \\ \sqrt{T} \exp\left(-\frac{1}{2}\widetilde{C}_{\text{SE}} T^{\frac{1}{d+1}}\right) & \text{if } T \geq \overline{T}_{\text{SE}}, \end{cases} \tag{31}$$

$$\sum_{t=1}^{T} \sigma(\boldsymbol{x}_t; \mathbf{X}_{t-1}) \leq \overline{T}_{\text{SE}} + (d+1) \left(\frac{\widetilde{C}_{\text{SE}}}{2}\right)^{-\frac{3d+3}{2}} \Gamma\left(\frac{3d+3}{2}\right), \tag{32}$$

*where $\widetilde{C}_{\text{SE}} = (6C_{\text{SE}})^{-\frac{1}{d+1}}$ and $\overline{T}_{\text{SE}} = \max\{\underline{T}_{\text{SE}}, \underline{T}_{\text{SE}}^{(\lambda)}, \lceil (d+1)^{d+1}/\widetilde{C}_{\text{SE}}^{d+1}\rceil + 1\}$ with $\underline{T}_{\text{SE}}^{(\lambda)} = \min\{T \in \mathbb{N}_+ \mid \forall t \geq T, t \exp(-\widetilde{C}_{\text{SE}} t^{1/(d+1)}) \leq \underline{\lambda}_{\text{SE}}^2\}$.*

- *For $k = k_{\text{Matérn}}$ with $\nu > 1/2$,*

$$\min_{t \in [T]} \sigma(\boldsymbol{x}_t; \mathbf{X}_{t-1}) \leq \begin{cases} 1 & \text{if } T < \overline{T}_{\text{Mat}}, \\ \widetilde{C}_{\text{Mat}}^{1/2} T^{-\frac{\nu}{d}} (\ln T)^{\frac{\nu}{d}} & \text{if } T \geq \overline{T}_{\text{Mat}}, \end{cases} \tag{33}$$

$$\sum_{t=1}^{T} \sigma(\boldsymbol{x}_t; \mathbf{X}_{t-1}) \leq \begin{cases} \overline{T}_{\text{Mat}} + \widetilde{C}_{\text{Mat}}^{1/2} \frac{d}{d-\nu} T^{\frac{d-\nu}{d}} (\ln T)^{\frac{\nu}{d}} & \text{if } d > \nu, \\ \overline{T}_{\text{Mat}} + \widetilde{C}_{\text{Mat}}^{1/2} (\ln T)^2 & \text{if } d = \nu, \\ \overline{T}_{\text{Mat}} + \widetilde{C}_{\text{Mat}}^{1/2} \frac{\Gamma(\frac{\nu}{d}+1)}{(\frac{\nu}{d}-1)^{\frac{\nu}{d}+1}} & \text{if } d < \nu, \end{cases} \tag{34}$$

*where $\widetilde{C}_{\text{Mat}} = \max\left\{1, \left(2 + \frac{2\nu}{d}\right)^{\frac{2\nu}{d}} (6C_{\text{Mat}})^{1+\frac{2\nu}{d}}\right\}$ and $\overline{T}_{\text{Mat}} = \max\{4, \underline{T}_{\text{Mat}}, \underline{T}_{\text{Mat}}^{(\lambda)}\}$ with $\underline{T}_{\text{Mat}}^{(\lambda)} = \min\{T \in \mathbb{N}_+ \mid \forall t \geq T, \widetilde{C}_{\text{Mat}} t^{-\frac{2\nu}{d}} (\ln t)^{\frac{2\nu}{d}} \leq \underline{\lambda}_{\text{Mat}}^2\}$.*

The upper bound in Lemma 8 depends on the quantities $\widetilde{C}_{\text{SE}}, \overline{T}_{\text{SE}}, \widetilde{C}_{\text{Mat}}, \overline{T}_{\text{Mat}} > 0$, which are related to the implied constants in the upper bounds of the MIG. However, under the fixed $d$, $\ell$, and $\nu$, $\widetilde{C}_{\text{SE}}$, $\overline{T}_{\text{SE}}, \widetilde{C}_{\text{Mat}}, \overline{T}_{\text{Mat}} > 0$ are also constants, which implies the conclusions of Lemma 3.

*Proof of Lemma 8.* When $k = k_{\text{SE}}$, we set $\lambda_t^2 = t \exp(-\widetilde{C}_{\text{SE}} t^{\frac{1}{d+1}})$, $\overline{T} = \overline{T}_{\text{SE}} := \max\{\underline{T}_{\text{SE}}, \underline{T}_{\text{SE}}^{(\lambda)}, \lceil (d+1)^{d+1}/\widetilde{C}_{\text{SE}}^{d+1}\rceil + 1\}$. From the definition of $\lambda_t^2$ and $\overline{T}_{\text{SE}}$, for any $t \geq \overline{T}_{\text{SE}}$, we have

$$\gamma_t(\lambda_t^2) \leq C_{\text{SE}} \left[\ln\left(\frac{t}{\lambda_t^2}\right)\right]^{d+1} \tag{35}$$

$$= C_{\text{SE}} \left[\ln \exp\left(\widetilde{C}_{\text{SE}} t^{\frac{1}{d+1}}\right)\right]^{d+1} \tag{36}$$

$$= C_{\text{SE}} \widetilde{C}_{\text{SE}}^{d+1} t. \tag{37}$$

Furthermore,

$$C_{\text{SE}} \widetilde{C}_{\text{SE}}^{d+1} t \leq \frac{t-1}{3} \Leftrightarrow \widetilde{C}_{\text{SE}}^{d+1} \leq \frac{t-1}{3C_{\text{SE}} t} \tag{38}$$

$$\Leftarrow \widetilde{C}_{\text{SE}}^{d+1} \leq \frac{1}{6C_{\text{SE}}} \tag{39}$$

$$\Leftrightarrow \widetilde{C}_{\text{SE}} \leq \left(\frac{1}{6C_{\text{SE}}}\right)^{\frac{1}{d+1}}, \tag{40}$$

where the second line follows from the inequality $t - 1 \geq t/2$ for all $t \geq \overline{T}_{\text{SE}} \geq 2$. By noting the definition of $\widetilde{C}_{\text{SE}}$, we conclude that $\forall t \geq \overline{T}_{\text{SE}}, \gamma_t(\lambda_t^2) \leq C_{\text{SE}} \widetilde{C}_{\text{SE}}^{d+1} t \leq \frac{t-1}{3}$ from the above inequalities, which implies that Lemmas 4 and 5 hold with $\lambda_t^2 = t \exp(-\widetilde{C}_{\text{SE}} t^{\frac{1}{d+1}})$ and $\overline{T} = \overline{T}_{\text{SE}}$. Eq. (31) directly follows from Lemma 4 using the fact $\sigma(\boldsymbol{x}_t; \mathbf{X}_{t-1}) \leq k(\boldsymbol{x}_t, \boldsymbol{x}_t) \leq 1$. As for Eq. (32), Lemma 5

implies

$$\sum_{t=1}^{T} \sigma(\boldsymbol{x}_t; \mathbf{X}_{t-1}) \leq \overline{T}_{\mathrm{SE}} + \sum_{t=\overline{T}_{\mathrm{SE}}}^{T} \lambda_t \tag{41}$$

$$\leq \overline{T}_{\mathrm{SE}} + \int_{\overline{T}_{\mathrm{SE}}-1}^{T} \sqrt{t} \exp\left(-\frac{1}{2}\widetilde{C}_{\mathrm{SE}} t^{\frac{1}{d+1}}\right) \mathrm{d}t \tag{42}$$

$$\leq \overline{T}_{\mathrm{SE}} + \int_{1}^{T} \sqrt{t} \exp\left(-\frac{1}{2}\widetilde{C}_{\mathrm{SE}} t^{\frac{1}{d+1}}\right) \mathrm{d}t, \tag{43}$$

where the second line follows from the fact that the function $g(t) := t \exp(-\widetilde{C}_{\mathrm{SE}} t^{1/(d+1)})$ is non-increasing for $t \geq \overline{T}_{\mathrm{SE}} - 1$. In fact, we have

$$g'(t) = \exp\left(-\widetilde{C}_{\mathrm{SE}} t^{\frac{1}{d+1}}\right)\left(1 - \frac{\widetilde{C}_{\mathrm{SE}}}{d+1} t^{\frac{1}{d+1}}\right), \tag{44}$$

which implies $g'(t) \leq 0$ for $t \geq \overline{T}_{\mathrm{SE}} - 1 \geq (d+1)^{d+1}/\widetilde{C}_{\mathrm{SE}}^{d+1}$. To bound the quantity $\int_{1}^{T} \sqrt{t} \exp\left(-\frac{1}{2}\widetilde{C}_{\mathrm{SE}} t^{\frac{1}{d+1}}\right) \mathrm{d}t$, we further derive the following upper bound with $C := \widetilde{C}_{\mathrm{SE}}/2 > 0$:

$$\int_{1}^{T} \sqrt{t} \exp\left(-C t^{\frac{1}{d+1}}\right) \mathrm{d}t = \int_{C}^{CT^{1/(d+1)}} \left(\frac{u}{C}\right)^{(d+1)/2} e^{-u}(d+1)\left(\frac{u}{C}\right)^d \frac{1}{C} \mathrm{d}u \quad (\because u = C t^{1/(d+1)}) \tag{45}$$

$$= (d+1)C^{-(3d+3)/2} \int_{C}^{CT^{1/(d+1)}} u^{(3d+1)/2} e^{-u} \mathrm{d}u \tag{46}$$

$$\leq (d+1)C^{-(3d+3)/2} \int_{0}^{\infty} u^{(3d+1)/2} e^{-u} \mathrm{d}u \tag{47}$$

$$= (d+1)C^{-(3d+3)/2} \Gamma\left(\frac{3d+3}{2}\right). \tag{48}$$

Next, when $k = k_{\mathrm{Matérn}}$, we set $\lambda_t^2 = \widetilde{C}_{\mathrm{Mat}} t^{-\frac{2\nu}{d}} (\ln t)^{\frac{2\nu}{d}}$ and $\overline{T} = \max\{4, \underline{T}_{\mathrm{Mat}}, \underline{T}_{\mathrm{Mat}}^{(\lambda)}\}$ with $\widetilde{C}_{\mathrm{Mat}} = \left(2 + \frac{2\nu}{d}\right)^{\frac{2\nu}{d}} (6 C_{\mathrm{Mat}})^{1+\frac{2\nu}{d}}$. Then, for any $t \geq \overline{T}_{\mathrm{Mat}}$, it holds that

$$\gamma_t(\lambda_t^2) \leq C_{\mathrm{Mat}}\left(\frac{t}{\lambda_t^2}\right)^{\frac{d}{2\nu+d}}\left[\ln\left(\frac{t}{\lambda_t^2}\right)\right]^{\frac{2\nu}{2\nu+d}} \tag{49}$$

$$= C_{\mathrm{Mat}}\widetilde{C}_{\mathrm{Mat}}^{-\frac{d}{2\nu+d}} t(\ln t)^{-\frac{2\nu}{2\nu+d}}\left[\ln\left(\widetilde{C}_{\mathrm{Mat}}^{-1} t^{\frac{d+2\nu}{d}}(\ln t)^{-\frac{2\nu}{d}}\right)\right]^{\frac{2\nu}{2\nu+d}} \tag{50}$$

$$= C_{\mathrm{Mat}}\widetilde{C}_{\mathrm{Mat}}^{-\frac{d}{2\nu+d}} t(\ln t)^{-\frac{2\nu}{2\nu+d}}\left[\ln\left(\widetilde{C}_{\mathrm{Mat}}^{-1}\right) + \frac{d+2\nu}{d}(\ln t) - \frac{2\nu}{d}(\ln\ln t)\right]^{\frac{2\nu}{2\nu+d}} \tag{51}$$

$$\leq C_{\mathrm{Mat}}\widetilde{C}_{\mathrm{Mat}}^{-\frac{d}{2\nu+d}} t(\ln t)^{-\frac{2\nu}{2\nu+d}}\left[\frac{2d+2\nu}{d}(\ln t)\right]^{\frac{2\nu}{2\nu+d}} \tag{52}$$

$$= C_{\mathrm{Mat}}\widetilde{C}_{\mathrm{Mat}}^{-\frac{d}{2\nu+d}} t\left(\frac{2d+2\nu}{d}\right)^{\frac{2\nu}{2\nu+d}}, \tag{53}$$

where the fourth line follows from $\widetilde{C}_{\mathrm{Mat}} \geq 1 \Rightarrow \widetilde{C}_{\mathrm{Mat}} \geq 1/t \Leftrightarrow \ln(\widetilde{C}_{\mathrm{Mat}}^{-1}) \leq \ln t$ for $t \geq 1$. Furthermore,

$$C_{\mathrm{Mat}}\widetilde{C}_{\mathrm{Mat}}^{-\frac{d}{2\nu+d}} t\left(\frac{2d+2\nu}{d}\right)^{\frac{2\nu}{2\nu+d}} \leq \frac{t-1}{3} \Leftrightarrow 3 C_{\mathrm{Mat}}\frac{t}{t-1}\left(\frac{2d+2\nu}{d}\right)^{\frac{2\nu}{2\nu+d}} \leq \widetilde{C}_{\mathrm{Mat}}^{\frac{d}{2\nu+d}} \tag{54}$$

$$\Leftrightarrow \left(\frac{3 C_{\mathrm{Mat}} t}{t-1}\right)^{1+\frac{2\nu}{d}}\left(2 + \frac{2\nu}{d}\right)^{\frac{2\nu}{d}} \leq \widetilde{C}_{\mathrm{Mat}} \tag{55}$$

$$\Leftarrow (6 C_{\mathrm{Mat}})^{1+\frac{2\nu}{d}}\left(2 + \frac{2\nu}{d}\right)^{\frac{2\nu}{d}} \leq \widetilde{C}_{\mathrm{Mat}}. \tag{56}$$

Combining the above inequalities, we can confirm $\forall t \geq \overline{T}_{\mathrm{Mat}}, \gamma_t(\lambda_t^2) \leq \frac{t-1}{3}$. Therefore, Lemmas 4 and 5 holds with $\lambda_t^2 = \widetilde{C}_{\mathrm{Mat}} t^{-\frac{2\nu}{d}} (\ln t)^{\frac{2\nu}{d}}$ and $\overline{T} = \overline{T}_{\mathrm{Mat}}$. Here, Eq. (33) is the direct consequence of Lemmas 4. As for Eq. (34), we have

$$\sum_{t=1}^{T} \sigma(\boldsymbol{x}_t; \mathbf{X}_{t-1}) \leq \overline{T}_{\mathrm{Mat}} + \sum_{t=\overline{T}_{\mathrm{Mat}}}^{T} \lambda_t \tag{57}$$

$$\leq \overline{T}_{\mathrm{Mat}} + \widetilde{C}_{\mathrm{Mat}}^{1/2} \int_{\overline{T}_{\mathrm{Mat}}-1}^{T} t^{-\frac{\nu}{d}} (\ln t)^{\frac{\nu}{d}} \mathrm{d}t \tag{58}$$

$$\leq \overline{T}_{\mathrm{Mat}} + \widetilde{C}_{\mathrm{Mat}}^{1/2} \int_{1}^{T} t^{-\frac{\nu}{d}} (\ln t)^{\frac{\nu}{d}} \mathrm{d}t, \tag{59}$$

where the second line follows from the fact that the function $g(t) := t^{-\frac{2\nu}{d}} (\ln t)^{\frac{2\nu}{d}}$ is non-increasing for $t \geq \overline{T}_{\mathrm{Mat}} - 1 \geq 3 > e$. Indeed, we have

$$g'(t) = \frac{2\nu}{d} t^{-\frac{2\nu}{d}-1} (\ln t)^{\frac{2\nu}{d}} \left( (\ln t)^{-1} - 1 \right), \tag{60}$$

which implies $g'(t) \leq 0$ for $t \geq e$. The desired results are obtained by bounding the quantity $\int_1^T t^{-\frac{\nu}{d}} (\ln t)^{\frac{\nu}{d}} \mathrm{d}t$ from above. When $d > \nu$, we have

$$\int_{1}^{T} t^{-\frac{\nu}{d}} (\ln t)^{\frac{\nu}{d}} \mathrm{d}t \leq (\ln T)^{\frac{\nu}{d}} \int_{1}^{T} t^{-\frac{\nu}{d}} \mathrm{d}t = (\ln T)^{\frac{\nu}{d}} \left[ \frac{d}{d-\nu} t^{\frac{d-\nu}{d}} \right]_1^T \leq \frac{d}{d-\nu} T^{\frac{d-\nu}{d}} (\ln T)^{\frac{\nu}{d}}. \tag{61}$$

When $d = \nu$,

$$\int_{1}^{T} t^{-\frac{\nu}{d}} (\ln t)^{\frac{\nu}{d}} \mathrm{d}t \leq (\ln T) \int_{1}^{T} t^{-1} \mathrm{d}t = (\ln T)^2. \tag{62}$$

When $d < \nu$, we have

$$\int_{1}^{T} t^{-\frac{\nu}{d}} (\ln t)^{\frac{\nu}{d}} \mathrm{d}t = \int_{0}^{\ln T} e^{-\left(\frac{\nu}{d}-1\right)u} u^{\frac{\nu}{d}} \mathrm{d}u \quad (\because u = \ln t) \tag{63}$$

$$\leq \int_{0}^{\infty} e^{-\left(\frac{\nu}{d}-1\right)u} u^{\frac{\nu}{d}} \mathrm{d}u \tag{64}$$

$$= \frac{\Gamma(\frac{\nu}{d}+1)}{\left(\frac{\nu}{d}-1\right)^{\frac{\nu}{d}+1}}, \tag{65}$$

where the last line follows from the standard property of Gamma function: $\int_0^\infty e^{-\lambda u} u^b \mathrm{d}u = \Gamma(b+1)/\lambda^{b+1}$ for any $\lambda > 0$ and $b > -1$ (e.g., Equation 6.1.1 in [Abramowitz and Stegun, 1968]). $\qquad \square$

## B  Detail of the Experiment

### B.1  Experimental settings for Figure 1

We give the detailed experimental settings used to plot Figure 1.

- **Objective function.** We define the true underlying objective function as $f(\cdot) = \sum_{m=1}^{50} c_m k(\boldsymbol{x}^{(m)}, \cdot)$, where $c_m \sim \mathrm{Uniform}([-1,1])$ and $\boldsymbol{x}^{(m)} \sim \mathrm{Uniform}([0,1]^2)$ are independently generated random variables. Note that by the definition of the RKHS, $f \in \mathcal{H}_k$ with $\|f\|_k = \sqrt{\sum_{m=1}^{50} \sum_{\widetilde{m}=1}^{50} c_m c_{\widetilde{m}} k(\boldsymbol{x}^{(m)}, \boldsymbol{x}^{(\widetilde{m})})}$.

- **Kernel.** We fix the lengthscale parameter $\ell$ as $\ell = 0.25$ in all experiments. We use the same kernel function in the GP-model used in the algorithms as the kernel leveraged to generate the objective function.

- **Other parameters.** We define the input domain $\mathcal{X}$ as the uniformly aligned $50 \times 50$ grid points on $[0,1]^2$. In all algorithms, we set the confidence width parameter $\beta^{1/2}$ to the exact RKHS norm. Furthermore, we set the initial batch size in PE and REDS as 5. Finally, we define the common initial point $\boldsymbol{x}_1$ for all the algorithms as the uniformly sampled point from $\mathcal{X}$.

In the above-described setting, we conduct experiments with 3000 different seeds.

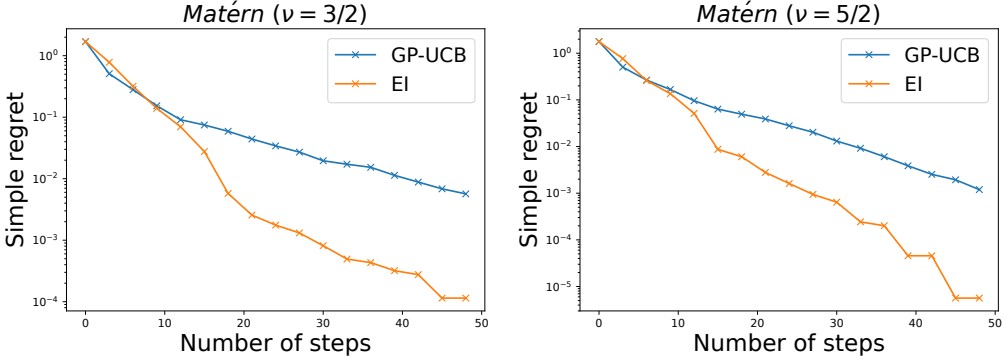

Figure 2: Comparison between GP-UCB and EI in simple regret minimization over 100 different seeds. We conduct experiments with Matérn kernel under $\nu = 3/2$ (left) and $\nu = 5/2$ (right).

## B.2 Comparison between EI and GP-UCB

Under the same setting as the previous subsection, we also compare GP-UCB's empirical performance with that of EI in simple regret minimization. Figure B.2 shows the results. We can confirm that, although GP-UCB achieves nearly-optimal worst-case regret, the empirical performance of GP-UCB is consistently worse than that of EI.

