# OpenReview forum: "Gaussian Process Upper Confidence Bound Achieves Nearly-Optimal Regret in Noise-Free Gaussian Process Bandits"
_NeurIPS.cc/2025/Conference — NeurIPS 2025 poster_

### Official Review · Reviewer_5t4c · 2025-06-16

**Clarity:** 3
**Significance:** 3
**Originality:** 2
**Rating:** 3
**Confidence:** 4

**Summary:**

This paper improves upon the regret analysis of [Iwazaki and Takeno, 2025a] for the Gaussian Process Upper Confidence Bound (GP-UCB) algorithm in the noise free GP bandits. In particular, the authors consider squared exponential and Matérn kernels, where they manage to improve the existing upper bounds on regret.

**Questions:**

I suggest extending the applications to other kernels, such as the Neural Tangent Kernel (NTK), which would broaden the impact and relevance of the proposed technique.

I find the presentation of the paper to be visually very similar to [Iwazaki and Takeno, 2025a]. I recommend revising the manuscript to better emphasize the originality of the contribution and to avoid potential plagiarism.

Additionally, in the contributions section, it is stated that GP-UCB achieves constant regret for $d > \nu$; however, according to Table 1 and as also indicated by Figure 1, it should be $d < \nu$. I assume this is a typo.

**Ethical Concerns:**

["NO or VERY MINOR ethics concerns only"]

**Final Justification:**

This paper improves upon the regret analysis of [Iwazaki and Takeno, 2025a] for the Gaussian Process Upper Confidence Bound (GP-UCB) algorithm in the noise-free GP bandit setting.

The achieved bounds are important for the field and were previously conjectured. However, I find the analysis to be incremental relative to prior work, and the presentation raises multiple plagiarism concerns. Therefore, I consider the paper to be borderline.

**Limitations:**

yes

**Paper Formatting Concerns:**

I do not have any concerns.

**Quality:**

2

**Strengths And Weaknesses:**

I find the problem addressed in the paper to be important, and the proposed method interesting, particularly because it enables the translation of noisy-regime guarantees to the noise-free setting. The paper is clearly written and generally easy to understand. Moreover, the technique has the potential to be generalized to a broader class of kernels.

However, I find the contribution, in its current form, insufficient for publication. While the refined analysis is valuable, it does not appear to be technically challenging and relies heavily on existing works. The scope of applications seems limited, and the difference from the prior work [Iwazaki and Takeno, 2025a] appears to be incremental.

In addition, I have concerns regarding the originality of the writing. Specifically, several parts of the manuscript—including the formulations of main theorems: Theorem 1 and Theorem 2, Table 1 and Table 2 (including their captions), Assumption 1, problem setting, and the subsections “Gaussian Process Model” and “Maximum Information Gain”—appear almost identical to the corresponding sections in [Iwazaki and Takeno, 2025a], without proper acknowledgment. This raises potential plagiarism concerns that should be carefully addressed.

---

> ### Author Rebuttal · Authors · 2025-07-30
>
> Thank you for taking the time to review our paper. Regarding the presentations that the reviewer pointed out, we promise to carefully revise them to avoid potential plagiarism. Here, please note that the revision of these presentations does not affect our contributions and can be conducted with minor revisions.
> However, we strongly disagree with the following reviewer's opinion that the current contributions of our paper are insufficient.
>
> **However, I find the contribution, in its current form, insufficient for publication. While the refined analysis is valuable, it does not appear to be technically challenging and relies heavily on existing works. The scope of applications seems limited, and the difference from the prior work [Iwazaki and Takeno, 2025a] appears to be incremental.**
>
> We clarify the reasons below.
>
> ### The difference from the prior work [Iwazaki and Takeno, 2025a] and the technical challenge.
> There are apparent differences between our work and [Iwazaki and Takeno, 2025a] in both the main results and the proof techniques. First, regarding the derived results: our main technical contribution, Lemma 3, is applicable to *any* algorithm, in contrast to the results in [Iwazaki and Takeno, 2025a], which are specific to the MVR algorithm. This algorithm-agnostic nature makes our contribution particularly important for the GP-bandits field, for several reasons:
>
> - As mentioned in Lines 152–159, our results address COLT open problems raised in [Vakili, 2022]. Specifically, [Vakili, 2022] conjectures that the input sequence of GP-UCB may satisfy the upper bounds provided in our Lemma 3 (See Chapter 4.4 in [Vakili, 2022]).  To the best of our knowledge, our work is the first to prove such a result.
> - In the setting under the squared exponential kernel and Matern with $d < \nu$, we obtain a strictly optimal constant $O(1)$ regret bound. Prior works such as [Iwazaki and Takeno, 2025a] and [Salgia et al., 2024] necessarily rely on phased elimination strategies using sequential non-adaptive algorithms. However, such approaches can only achieve polylogarithmic regret even under highly smooth objective functions. In contrast, our algorithm-independent upper bound enables us to derive regret without relying on any phased elimination structure and attains a constant $O(1)$ regret. To our knowledge, no existing work in the GP-bandit literature provides constant or strictly optimal regret bounds, marking a significant and novel contribution beyond [Iwazaki and Takeno, 2025a].
> - Our result enables the extension of many existing analyses in the noisy setting, especially those based on[Srinivas et al., 2010], which plays a central role in GP-bandit literature, to stronger guarantees in the noiseless setting. This point is made explicit in Lines 160–169 and elaborated further in the latter section of this rebuttal (“Scope of Applications”).
>
> Regarding proof techniques, there is a distinct departure in methodology. The proof in [Iwazaki and Takeno, 2025a] relies on explicitly upper-bounding the sum of the posterior variance (not the standard deviations) using the information gain, extending the approach of [Srinivas et al., 2010], in combination with the elliptical potential count lemma (Lemma 6 in our paper). In contrast, as described in Lines 199–232, our proof avoids explicit reliance on the information gain-based upper bound of the sum of posterior variance. Instead, we combine the elliptical potential count lemma with the monotonicity of the posterior standard deviations with respect to the noise parameter and the training data, resulting in a fundamentally different and more direct application of the elliptical potential count lemma.
>
> While both our work and [Iwazaki and Takeno, 2025a] employ the elliptical potential count lemma, the way it is utilized differs significantly. Notably, the elliptical potential count lemma itself was initially applied in [Flynn & Reeb, 2025] to analyze GP-UCB. However, as stated in Chapter 3 of [Iwazaki and Takeno, 2025a] and shown in [Flynn & Reeb, 2025], combining this lemma with the standard information gain-based upper bound of the posterior variance by [Srinivas et al., 2010] leads to strictly suboptimal results unless applied to particular algorithms such as MVR. This makes it technically non-trivial to use the elliptical potential count lemma to achieve near-optimal results in noise-free GP-bandit fields. We overcome this challenge by developing a new proof strategy that bypasses the explicit information gain-based bound of the posterior variance in [Srinivas et al., 2010] entirely and instead relying on the repeated application of the elliptical potential count lemma with the monotonicity with respect to the noise parameter and training input.
>
> ### The scope of applications
>
> As detailed in Lines 160–169, our Lemma 3 enables a extension of regret analyses in noisy settings -often characterized by terms involving either $\sum_{t=1}^T \sigma(x_t; X_{t-1})$ or $\min_{t \in [T]} \sigma(x_t; X_{t-1})$- to the noiseless setting, without requiring algorithm-specific structure.
> This is possible because, in many existing results, the dominant terms in the regret or performance bounds are quantified by these quantities. Specifically, as described in
> Below, we elaborate on the examples cited in the main text:
>
> - **Bayesian GP-UCB**: As in \[Srinivas et al., 2010, Theorems 1 and 2], one obtains regret bounds of the form
>   $R_T \leq O(\sqrt{\beta_T} \sum_{t=1}^T \sigma(x_t; X_{t-1}))$, with $\sqrt{\beta_T} = O(\ln T)$.
>
> - **Bayesian GP-TS**: As shown in [Russo & Roy, 2014, Chapter 6.3 and Proposition 5],
>   $R_T \leq O(\sqrt{\beta_T} \sum_{t=1}^T \sigma(x_t; X_{t-1}))$, where again $\sqrt{\beta_T} = O(\ln T)$.
>
> - **Contextual GP bandits**: From \[Krause & Ong, 2011, Lemma 4.1], CGP-UCB algorithm satisfies $R_T \leq O(\sqrt{\beta_T} \sum_{t=1}^T \sigma(x_t; X_{t-1}))$. In frequentist settings, based on Lemma 7 of our paper, this assumes a fixed-width confidence bound $\sqrt{\beta_t} = B$. Furthermore, the input space $\mathcal{X}$ is defined as the product of the context and decision spaces.
>
> - **Level-set estimation**: By combining Lemmas 5 and 6 of [Gotovos et al., 2013], the stopping time required to identify an $\epsilon$-accurate solution using the straddle algorithm is bounded by the smallest $T$ satisfying $2 \sqrt{\beta_T} \min_{t \in [T]} \sigma(x_t; X_{t-1}) \leq \epsilon$.
>
> - **Multi-objective optimization**: As in [Zuluaga et al., 2016, Section 4.2], the number of queries required to identify an $\epsilon$-Pareto set using $\epsilon$-PAL is upper bounded by the smallest $T$ satisfying
>   $2m \sqrt{\beta_T} \max_{i \in [m]} \min_{t \in [T]} \sigma_i(x_t; X_{t-1}) \leq \epsilon$,
>   where $\sqrt{\beta_t} = B$ (by Lemma 7 of our paper), $m$ is the number of objective functions, and $\sigma_i(\cdot)$ is the posterior standard deviation of the $i$-th GP.
>
> - **Robust optimization**: From Eq.(42) in [Bogunovic et al., 2018], the robust regret of Stable-OPT is given by
>   $O(\sqrt{\beta_T} \sum_{t=1}^T \sigma(x_t; X_{t-1}) / T)$. Here, we assume $\sqrt{\beta_t} = B$ based on Lemma 7 of our paper.
>
> These are merely examples, and we think that many other results based on the analysis by [Srinivas et al., 2010] for the noisy GP-bandit setting can also be extended to the noise-free setting by using our Lemma 3.
>
> Finally, we would like to add a note on the applications of the noise-free setting itself. Applications of the noise-free setting are not limited to Gaussian process bandits but have been extensively studied in related fields such as Efficient Global Optimization and Bayesian optimization. In particular, in engineering and scientific domains, dealing with noiseless black-box functions—such as in computer simulations and system tuning—is a critical problem. For example, relevant applications and studies are comprehensively referred to in Chapter 1 of [De Freitas et al., 2012] and in the tutorial paper by [Frazier,2018]. (Notably, the tutorial paper by [Frazier,2018] treats the noiseless setting as the "standard" problem.)
>
> Based on the above clarifications, we sincerely hope that the reviewer reconsider the current score or clarify the unresolved concerns in detail. We believe that the results presented in our paper are of significant importance to the field, and we strongly feel that the paper should not be rejected due to misunderstandings.
>
> - Srinivas, N., Krause, A., Kakade, S., & Seeger, M. (2010). Gaussian Process Optimization in the Bandit Setting: No Regret and Experimental Design.
>
> - Krause, A., & Ong, C. (2011). Contextual Gaussian process bandit optimization.
>
> - De Freitas, N., Smola, A. J., & Zoghi, M. (2012). Exponential regret bounds for Gaussian process bandits with deterministic observations.
>
> - Gotovos, A., Casati, N., Hitz, G., & Krause, A. (2013). Active learning for level set estimation.
>
> - Russo, D., & Van Roy, B. (2014). Learning to optimize via posterior sampling.
>
> - Zuluaga, M., Krause, A., & Püschel, M. (2016). e-pal: An active learning approach to the multi-objective optimization problem.
>
> - Bogunovic, I., Scarlett, J., Jegelka, S., & Cevher, V. (2018). Adversarially robust optimization with Gaussian processes.
>
> - Frazier, P. I. (2018). A Tutorial on Bayesian Optimization.
>
> - Vakili, S. (2022). Open problem: Regret bounds for noise-free kernel-based bandits.
>
> - Li, Z., & Scarlett, J. (2024). Regret Bounds for Noise-Free Cascaded Kernelized Bandits.
>
> - Salgia, S., Vakili, S., & Zhao, Q. (2024). Random Exploration in Bayesian Optimization: Order-Optimal Regret and Computational Efficiency.
>
> - Flynn, H., & Reeb, D. (2025). Tighter Confidence Bounds for Sequential Kernel Regression.

---

> > ### Comment · Reviewer_5t4c · 2025-08-03
> >
> > Thank you for the clarification!
> >
> > I have re-evaluated the importance of the contribution and increased the score.

---

> > ### Comment · Reviewer_5t4c · 2025-08-03
> >
> > Thank you for the clarification!
> >
> > I have re-evaluated the importance of the contribution and increased the score.

---

> > > ### Author Response · Authors · 2025-08-03
> > >
> > > We thank the reviewer for your reply. We will carefully incorporate your feedback about the presentation in the revision. Thank you again.

---

> > > > ### Comment · Reviewer_5t4c · 2025-08-05
> > > > **Score Clarification**
> > > >
> > > > I have realized that the new score is not visible to the authors and I didn't communicate it properly. Thus, I wish to clarify that I increased my score to a borderline reject, as I found my previous assessment to be too harsh in light of the importance of the results, as explained to me during the rebuttal.
> > > >
> > > > However, I still find the analysis to be incremental relative to prior work, and the presentation raises multiple plagiarism concerns. Therefore, I consider the paper to be borderline.
> > > >
> > > > If other reviewers are certain about accepting the paper, I wouldn’t object, but I would not support its acceptance myself for the reasons described above.

---

> > > > > ### Author Response · Authors · 2025-08-06
> > > > >
> > > > > Thank you for your score clarification.
> > > > >
> > > > > > *I still find the analysis to be incremental relative to prior work...*
> > > > >
> > > > > As stated in our first rebuttal comments with reasons, we respectfully disagree with the view that our analysis lacks technical novelty. At this point, it is unclear to us whether this concern arises from a misunderstanding or from a different perspective in evaluating our analysis. We would greatly appreciate it if you could clarify the specific reasons why you found our analysis to be incremental. We are fully prepared to engage in discussion until the extended final Author-Reviewer deadline.
> > > > >
> > > > > > *...and the presentation raises multiple plagiarism concerns*
> > > > >
> > > > > We sincerely appreciate you pointing out the presentation where our attention has been insufficient. We take this issue very seriously and will ensure that the final version strictly adheres to the NeurIPS code of ethics to avoid any possibility of plagiarism. Unfortunately, NeurIPS does not allow sharing a revised PDF paper during the review process. However, if the reviewer considers it would be helpful to address the concerns, we would be happy to share a current revised presentation of the relevant parts via "official comment".

---

> > > > > > ### Comment · Reviewer_5t4c · 2025-08-08
> > > > > >
> > > > > > Thanks for your reply. I am happy to clarify my concerns.
> > > > > >
> > > > > > According to your clarification of the proof techniques:
> > > > > >
> > > > > > **Instead, we combine the elliptical potential count lemma with the monotonicity of the posterior standard deviations with respect to the noise parameter and the training data, resulting in a fundamentally different and more direct application of the elliptical potential count lemma.**
> > > > > >
> > > > > > To my understanding, your proof has two main components: monotonicity and the potential count lemma. The first is presented without proof and is considered trivial in your manuscript. The potential count lemma has been used before in the same context. In my view, this represents a refinement of an existing proof technique and therefore offers only limited novelty.
> > > > > >
> > > > > > Regarding plagiarism, it is a scientific misconduct, which makes it ethically difficult for me to support such a paper. I do not doubt that the manuscript could be revised, but I leave the decision to the AC.
> > > > > >
> > > > > > I hope this clarifies my score.

---

> > > > > > > ### Author Response · Authors · 2025-08-08
> > > > > > >
> > > > > > > We appreciate the reviewer’s clarification. Regarding the concerns about the novelty, we hope the response below helps to address the reviewer’s concerns.
> > > > > > >
> > > > > > > First, we would like to clarify that we do not claim novelty in the use of the elliptical potential count (EPC) lemma and monotonicity in the noiseless setting. As correctly noted by the reviewer, there is no novelty in this aspect. As stated in our initial rebuttal, the novelty of our analysis lies in **how** the EPC lemma is utilized.
> > > > > > >
> > > > > > > To our knowledge, almost all analyses in this field over the past $15$ years have followed the proof technique introduced by \[Srinivas et al., 2010], which upper bounds the cumulative posterior standard deviation via an maximum information gain explicitly. Even prior works that consider noiseless settings, such as [Flynn & Reeb, 2025; Iwazaki and Takeno, 2025a], have applied the EPC lemma within this framework of [Srinivas et al., 2010].
> > > > > > >
> > > > > > > In contrast, our analysis departs from this tradition. We propose a direct, iterative application of the EPC lemma, which is significantly different from the proof technique in [Srinivas et al., 2010]. Our results demonstrate that this approach yields stronger algorithm-independent bounds, which is a new insight. We believe that this is a novel contribution, particularly for theoretical researchers familiar with the line of analysis based on [Srinivas et al., 2010].
> > > > > > > Furthermore, to our knowledge, the application of the EPC lemma to GP-bandit problems has only begun to be explored in the past one year. Its potential in GP-bandit field remains uncertain. Therefore, we believe that the use of the EPC lemma itself does not diminish the novelty of our work. Rather, our contribution lies in developing a new proof strategy that leverages EPC outside the traditional [Srinivas et al., 2010] framework.
> > > > > > >
> > > > > > >
> > > > > > > We sincerely appreciate your continued efforts to the review of our paper. Please do not hesitate to reach out with further questions or points of clarification.

---

> > > > > > > > ### Author Response · Authors · 2025-08-08
> > > > > > > > **Replay for the presentation concerns**
> > > > > > > >
> > > > > > > > We appreciate your pointing out potential plagiarism issues, which we had unfortunately overlooked and must take responsibility for as the author.
> > > > > > > > Before the Area Chair makes a final judgment, we would be grateful if we could share our current understanding of the matter.
> > > > > > > >
> > > > > > > > Upon revisiting the reviewer’s comments and carefully reviewing the presentation in [Iwazaki & Takeno, 2025], we agree that the presentation issues of our paper stem from the similarity of the following parts in our submission:
> > > > > > > >
> > > > > > > > 1. The presentation of the general problem setting in Section 2, and the presentation of the Gaussian process model and the maximum information gain.
> > > > > > > > 2. The way theorems were written.
> > > > > > > > 3. The description of the caption and the insufficient citation in the table.
> > > > > > > >
> > > > > > > > We sincerely apologize for not being sufficiently careful in citing prior work properly and structuring our presentation to avoid similarities with the above points in existing work. That said, we would like to clarify that these parts do not affect the originality or contributions of our paper. While we fully agree that these issues should be revised to avoid any resemblance to the presentations in prior work, we believe that the current presentation regarding (1) and (2) also falls within the standard terminology and descriptions commonly used in the GP-bandit and mathematical literature, not limited to [Iwazaki & Takeno, 2025].
> > > > > > > >
> > > > > > > > Although we agree that modifications of the presentation are necessary, we believe that the current paper is unlikely to exhibit a level of similarity to existing works that would meet the general criteria for an academic plagiarism case. (Please understand that we cannot provide quantitative evidence of similarity using online plagiarism checkers since the submitted paper is confidential.) Therefore, we believe that the presentation issues of our paper fall within the scope of camera-ready revisions.

---

### Official Review · Reviewer_TJZH · 2025-06-30

**Clarity:** 3
**Significance:** 3
**Originality:** 3
**Rating:** 5
**Confidence:** 3

**Summary:**

**Setting**
This paper considers the noise-free Gaussian Process Bandit setting, where in each round, an arm $x_t$ is selected and the corresponding reward is $f(x_t)$. The function $f(\cdot)$ is unknown, and only function values can be observed. Since the observation is exactly the function value, this constitutes a noise-free scenario. Two performance measures are considered: one is the standard cumulative regret  and the other is the simple regret of the final round. The paper analyzes two types of kernel functions: the squared exponential (SE) kernel and the Matérn kernel.

**Approach**
The paper does not propose a new algorithm, but instead provides a refined theoretical analysis for the classical GP-UCB method. In the case of the squared exponential kernel, it improves the previously known regret bound from $\mathcal{O}(\ln T)$ to $\mathcal{O}(1)$. For the Matérn kernel, the analysis also yields improved regret bounds. The theoretical improvement mainly comes from a tighter control of the posterior standard deviations, specifically the term $\sum_{t=1}^T \sigma(x_t| X_{t-1})$.

**Questions:**

I'm confused about why GP-UCB can achieve exponentially decaying simple regret, since the UCB strategy is fundamentally designed for balancing exploration and exploitation, rather than for pure exploration. In the context of pure exploration or Best Arm Identification (BAI), wouldn't existing BAI strategies be more suitable or more effective than UCB? Are there BAI algorithms that can achieve even better convergence rates? Or are UCB-based methods and pure exploration algorithms essentially equivalent in this noise-free setting?

**Ethical Concerns:**

["NO or VERY MINOR ethics concerns only"]

**Final Justification:**

This paper provides a clearly improved analysis and better theoretical guarantees for GP-UCB.
My main concern was whether a regret minimization algorithm can indeed perform well in terms of simple regret (i.e., in pure exploration settings). In their rebuttal, the authors addressed this concern by presenting experimental results showing that regret minimization algorithms still underperform compared to pure exploration methods for minimizing simple regret. This resolves my concern, and I therefore recommend acceptance.

**Limitations:**

The main limitation lies in the fact that the analysis is restricted to only two kernel functions, and there is no algorithmic innovation throughout the paper. The contribution is primarily a refined theoretical analysis of an existing algorithm rather than the proposal of a new method. But I believe the contribution of this work is sufficient.

**Quality:**

3

**Strengths And Weaknesses:**

**Strength**:

- The theoretical analysis is very solid, with clearly stated improvements and contributions.
- A key strength of the paper is its effective exploitation of structural properties in the noise-free setting. Previous works that considered the noise-free case typically focused on refining the estimation error $\beta$, which, in the absence of noise, becomes a constant instead of a time-dependent term, thereby improving the regret order from a factor of $\log T$. This paper goes further by observing that the cumulative posterior standard deviation $\sum_{t=1}^T \sigma(x_t, X_{t-1})$ can also be more tightly controlled in the noise-free setting. This observation is very reasonable: in the noise-free case, once an arm $x$ is queried, its uncertainty $\sigma(x)$ drops immediately to zero. Therefore, each arm only needs to be explored once to determine its value, significantly reducing the amount of exploration required.

**Weakness**:

I did not find any obvious weaknesses. If the theoretical results hold as claimed, I believe this is a highly meaningful and valuable contribution.

---

> ### Author Rebuttal · Authors · 2025-07-30
>
> Thank you for your overall positive assessments.
>
> **Q. I'm confused about why GP-UCB can achieve exponentially decaying simple regret, since the UCB strategy is fundamentally designed for balancing exploration and exploitation, rather than for pure exploration. In the context of pure exploration or Best Arm Identification (BAI), wouldn't existing BAI strategies be more suitable or more effective than UCB? Are there BAI algorithms that can achieve even better convergence rates? Or are UCB-based methods and pure exploration algorithms essentially equivalent in this noise-free setting?**
>
> It is an insightful question. At least in the worst-case regret under Matern kernels, GP-UCB also achieves nearly-optimal regret, which indicates that we cannot obtain strictly superior BAI or pure exploration algorithm to GP-UCB in the worst-case sense. However, as far as we know, the empirical performance of GP-UCB for the simple regret tend to be worse than those of the algorithms tailored for simple regret, such as the expected improvement (EI) algorithm.
> Actually, we conducted an experiment under the same setting as Figure 1 in the main text, and observed that EI outperforms GP-UCB in terms of simple regret. The results are summarized in the table below.
>
> | | t=10 | t=20 | t=30 | t=40 | t=50 |
> | --- | --- | --- | --- | --- | --- |
> | GP-UCB under Mat'ern 5/2 | 0.1319 | 0.04241 | 0.0131 | 0.00316 | 0.0006518 |
> | EI under Mat'ern 5/2 | 0.09675 | 0.003263 | 0.0006412 | 4.563e-05 | 5.648e-06 |
> | GP-UCB under Mat'ern 3/2 | 0.125 | 0.05118 | 0.01957 | 0.01051 | 0.005216 |
> | EI under Mat'ern 3/2 | 0.1055 | 0.003216 | 0.0008133 | 0.0003194 | 2.152e-05 |
>
> It remains unclear whether this performance gap arises from differences in constants or logarithmic factors in the worst-case regret bounds, or whether it reflects a more fundamental mismatch between worst-case analysis and practical performance. In the latter case, instance-dependent analyses—-such as those proposed in [Shekhar & Javidi, 2022] for GP bandits—-may provide deeper insights into the differences between GP-UCB and other algorithms specifically designed for BAI and pure exploration. To the best of our knowledge, in the context of GP bandits, there has been little evidence suggesting a fundamental difference in the analysis of cumulative regret versus simple regret. Therefore, we believe that the question raised by the reviewer points to an important open problem in the field.
> We are happy to include a discussion of these points in Section 4 in the revision.
>
> - Shekhar, S., & Javidi, T. (2022, June). Instance dependent regret analysis of kernelized bandits. In International Conference on Machine Learning (pp. 19747-19772). PMLR.

---

> > ### Comment · Reviewer_TJZH · 2025-08-05
> >
> > Thanks for the response. My concern has been resolved. I will maintain my score.

---

### Official Review · Reviewer_UxhJ · 2025-07-02

**Clarity:** 4
**Significance:** 4
**Originality:** 3
**Rating:** 5
**Confidence:** 4

**Summary:**

This paper resolves a significant gap between the empirical performance and theoretical understanding of the Gaussian Process Upper Confidence Bound (GP-UCB) algorithm for noise-free bandit optimization. Although GP-UCB is a popular and practically effective adaptive algorithm, its existing regret guarantees were suboptimal. This work provides a new, refined analysis that proves GP-UCB achieves nearly-optimal regret, aligning its theoretical guarantees with its strong empirical results. The core technical contribution is a new, algorithm-independent upper bound on the sum of posterior standard deviations, derived by connecting analysis techniques from the noisy GP setting (specifically, the Maximum Information Gain) to the noise-free case.

The new analysis establishes significantly tighter regret bounds that match known lower bounds up to polylogarithmic factors. For cumulative regret, the paper shows that GP-UCB achieves constant $O(1)$ regret for the squared exponential (SE) kernel, and different near-optimal rate for the Matérn kernel depending on the relationship of the input dimension $d$ and the smoothness parameter $v$ (summarized in Table 1). For simple regret, the analysis provides rates of $O(\sqrt{T} exp(-\frac{1}{2} CT^{\frac{1}{d+1}})$ for the SE kernel, and $\tilde{O}(T^{-\frac{v}{d}})$ for the Matérn kernel (summarized in Table 2). Based on the summary in the literature review, these two results advance the state-of-the-art regret bounds and are near-optimal compared to the best-known lower bounds.
​
Because the main technical lemmas are algorithm-independent, this work not only solidifies the theoretical foundation of GP-UCB but also provides a powerful new tool that could be used to improve the analysis of many other GP-based bandit algorithms.

I have checked the proofs in the main paper and the proofs are sound to me. The new analysis on the posterior standard deviation is also neat and new to me. I think the paper is a solid contribution to the analysis of GP-UCB regret with a simple yet new idea.

**Questions:**

Nothing specific in my mind. The paper is very clear and well-structured. It would be great if the authors can also briefly share the results of applying the same idea to RBF and linear kernels, and if this idea advances the best-known regret bounds for these kernels.

**Ethical Concerns:**

["NO or VERY MINOR ethics concerns only"]

**Final Justification:**

I have read the authors' responses and I keep my assessment.

**Limitations:**

Yes

**Quality:**

4

**Strengths And Weaknesses:**

Strengths:
- Very clear presentation and neat idea.
- Solid improvement of the regret bound in both the simple and cumulative settings.
- Very clear summary of the literature

Neutral feedback:
- As the authors mention, the idea can also be applied to other kernels. It would be great if the results of other kernels can also be shown or briefly summarized in the paper.

---

> ### Author Rebuttal · Authors · 2025-07-30
>
> Thank you for your overall positive assessments.
>
> **Q. It would be great if the authors could also briefly share the results of applying the same idea to RBF and linear kernels, and if this idea advances the best-known regret bounds for these kernels.**
>
> For the linear kernel, we can obtain $\min \sigma(x_t; X_{t-1}) = O(\sqrt{T} \exp(-\frac{T}{2d}))$ and
> $\sum_{t=1}^T \sigma(x_t; X_{t-1}) = O(1)$, since we can set $\lambda_t = \Theta(\sqrt{T} \exp(-\frac{T}{2d}))$ from $\gamma_T(\lambda^2) = O(d \ln (T/\lambda^2))$ [Srinivas et al., 2010]. However, under the linear kernel with noiseless setting, if the input sequence is not redundant, we can immediately obtain $\min \sigma(x_t; X_{t-1}) = 0$ for $t > d$ and $\sum_{t=1}^T \sigma(x_t; X_{t-1}) = O(1)$ without using our lemmas. Therefore, we believe that our results do not provide any advanced insight under the linear kernel so far.
> The results for the RBF kernel are the same as the results for the SE kernel, as long as the reviewer adopts the standard definition of the RBF kernel, which is the same as the SE kernel.
>
> - Srinivas, N., Krause, A., Kakade, S., & Seeger, M (2010). Gaussian Process Optimization in the Bandit Setting: No Regret and Experimental Design.

---

> > ### Comment · Reviewer_UxhJ · 2025-08-04
> >
> > Thank you for your response. I have read the responses and I keep my current assessment.

---

### Decision · Program_Chairs · 2025-09-17

**Decision:**

Accept (poster)

**Comment:**

This primarily theoretically oriented paper makes important contributions in the fundamental understanding of Gaussian process (GP) based bandit algorithms that rely on noiseless function value observations, as opposed to the classically studied noisy observation setting. Specifically, it develops refined bounds on the cumulative and minimum posterior variances across a sequence of actions, under commonly used kernels. These are then used to resolve open questions about the cumulative regret performance of the GP-Upper Confidence Bound (GP-UCB) algorithm in the noiseless setting, by showing that it achieves nearly optimal regret rates for commonly used kernels such as the Squared Exponential and Matérn kernels. As such, the results about bounding the posterior variance are of a plug-and-play form and hold promise for endowing any existing confidence-bound based regret analyses with improvements in regret rates.

The referees all agree that the central contribution of the paper -- in developing improved bounds for posterior variances -- is of significant value in the black box GP optimization literature. Though one of the referees, with a borderline score, raised concerns about the overall quantum of novelty of the submission, the authors responded with detailed clarifications about the scope and importance of the contribution. As a result, I am of the view that the significance of the technical contribution is no longer a lingering concern.

In view of the strengths of the paper as brought out in the reviews and ensuing discussions, I am happy to recommend that the paper be accepted. I urge the authors to consider the suggestions of the referees to improve the clarity of presentation in a revised version.